# A randomized controlled trial for response of microbiome network to exercise and diet intervention in patients with nonalcoholic fatty liver disease

Runtan Cheng[1,2,3,13], Lu Wang[4,13], Shenglong Le[1,5], Yifan Yang [1,3], Can Zhao[6], Xiangqi Zhang[1,3], Xin Yang [3], Ting Xu[3], Leiting Xu[1,7], Petri Wiklund[1,2], Jun Ge[1,8], Dajiang Lu[2,9], Chenhong Zhang [3,10✉], Luonan Chen[4,11,12✉] & Sulin Cheng [1,2,5✉]

Exercise and diet are treatments for nonalcoholic fatty liver disease (NAFLD) and pre-diabetes, however, how exercise and diet interventions impact gut microbiota in patients is incompletely understood. We previously reported a 8.6-month, four-arm (Aerobic exercise, $n = 29$; Diet, $n = 28$; Aerobic exercise + Diet, $n = 29$; No intervention, $n = 29$) randomized, singe blinded (for researchers), and controlled intervention in patients with NAFLD and prediabetes to assess the effect of interventions on the primary outcomes of liver fat content and glucose metabolism. Here we report the third primary outcome of the trial—gut microbiota composition—in participants who completed the trial (22 in Aerobic exercise, 22 in Diet, 23 in Aerobic exercise + Diet, 18 in No Intervention). We show that combined aerobic exercise and diet intervention are associated with diversified and stabilized keystone taxa, while exercise and diet interventions alone increase network connectivity and robustness between taxa. No adverse effects were observed with the interventions. In addition, in exploratory ad-hoc analyses we find that not all subjects responded to the intervention in a similar manner, when using differentially altered gut microbe amplicon sequence variants abundance to classify the responders and low/non-responders. A personalized gut microbial network at baseline could predict the individual responses in liver fat to exercise intervention. Our findings suggest an avenue for developing personalized intervention strategies for treatment of NAFLD based on host-gut microbiome ecosystem interactions, however, future studies with large sample size are needed to validate these discoveries. The Trial Registration Number is ISRCTN 42622771.

A full list of author affiliations appears at the end of the paper.

Nonalcoholic fatty liver disease (NAFLD) is the most common chronic liver disease with prevalence estimates ranging from 25% to 45% in worldwide[1,2]. NAFLD is closely associated with type 2 diabetes (T2D)[3], most likely via a common pathophysiological mechanism - insulin resistance[4]. Therefore, interventions targeting the synergistically pathogenic mediator of NAFLD and diabetes are likely to be a rational approach in the prevention and treatment of the comorbidity condition.

The gut microbiota, which contributes to the metabolic health of the human host, may work as one of the targets for the treatment of NAFLD. Emerging data demonstrate that dysbiosis of the gut microbiota is linked with NAFLD and T2D[5,6]. Although the exact mechanism(s) require further elucidation, inflammation, damage to the intestinal membrane, and translocation of bacteria have all been suggested[7]. Drugs with antifibrotic or anti-inflammatory treatments for various stages of NAFLD in human trials is pending[8]. Currently, increased physical activity and dietary modifications are the only effective therapeutic options for NAFLD management, and the mechanisms of these interventions have been associated with modulation of gut microbiota and its metabolites[9]. For example, a low-carbohydrate diet (LCD) has been reported to improve fatty liver metabolism and to promote rapid shifts of the gut microbiota composition in NAFLD patients[10]. Regular aerobic exercise has been shown to reduce hepatic fat in obesity[11,12], and some studies have shown that exercise increases gut microbial diversity[13] and alters the composition and functional capacity of gut microbiota[14,15]. However, the underlying mechanism of the beneficial effects of exercise and diet on NAFLD, as well as the related metabolic disorders through regulation of the gut microbiota, still remain to be elucidated.

One of the challenges in constructing the relationship between the gut microbiota and improvement of NAFLD during exercise and/or dietary interventions is that not all patients respond to the intervention in a similar manner. Earlier studies have shown that there was an individual variation in response to exercise, with some subjects experiencing greater improvement than others[16]. The low/non-responders could be up to 50% in terms of change of gut microbiota after exercise intervention[17]. Some researchers have suggested that the variation in responsiveness might be due to genotypic and phenotypic factors[18,19]. Thus, large inter-individual variance may mask the changes of the microbiome in studies using traditional statistical analytical methods. Currently, it is widely accepted that, in the microbial community, the keystone taxa are drivers of microbiome structure and function, and in particular, their interaction network, which plays an important role in microbial functions and disease progression[20]. Therefore, to establish an effective intervention strategy, it may be worthwhile to analyze the microbiota correlation network at the microbial community level, while assessing the physiological mechanisms at the individual level and further exploring individual microbial networks that underlie the differences between responders and low/non-responders of various interventions.

In the present study, we analyzed the composition and metabolic pathways of the gut microbiota, constructed a co-occurrence network at the population level, and developed personalized gut microbial networks. This individual network enabled us to identify the microbial signature and interaction of taxa at a personal resolution, and to further differentiate responders from low/non-responders after intervention. Taken together, our study with relatively long-term intervention provides both statistical and sample-specific network insights into the complex gut microbial ecosystem in patients with NAFLD and glucose metabolism impairment.

## Results

**Participant characteristics and gut microbiota research design.** This study was an 8.6-month, four-arm, randomized trial (Fig. 1a, Fig. S1) and participant characteristics have been reported in our previous publication[21]. Briefly, 115 participants were recruited from 7 health clinical service centers in the Shanghai Yangpu district. They were randomized into four groups: aerobic exercise intervention (AEX, $n = 29$), fiber-enriched low-carbohydrate diet intervention (Diet, $n = 28$), aerobic exercise combined with diet intervention (AED, $n = 29$) and no intervention without guided exercise and dietary intake (NI, $n = 29$). Of all participants, 85 individuals completed the intervention trial.

For the primary outcome of liver fat content, we previously found that hepatic fat content (HFC) was significantly reduced in the exercise AEx (–24.4%), diet (–23.2%), and AED (–47.9%) groups by contrast to the 20.9% increase in the NI group ($p < 0.001$ for all) after intervention[21]. Of note, 91% of the subjects in the AED group decreased their HFC, and the corresponding figures were 68% in the AEx group, and 86% in the Diet group. In contrast, 72% in the NI group increased their HFC during the intervention period. However, for the primary outcome of glucose metabolism, no significant remission or progression of prediabetes was found between the intervention and NI groups based on the glycated hemoglobin A1c (HbA1c)[21]. These results indicated that our intervention was effective mainly for HFC reduction.

We found that not all the subjects responded to the intervention in a similar manner. Therefore, we stratified the participants into responders (HFC decreased more than 5%) and low/non-responders (HFC decrease less than 5% or increased) (Fig. 1b). For the primary outcome of gut microbiota composition, seventy-six subjects provided paired stool samples at baseline and after the intervention for 16 s rRNA gene sequencing. The demographics and clinical variables of those who have had gut microbiota results are presented to Table S1. In addition, to better understand the intervention impact on the function of microbiome, we selected a subset of the cohort ($n = 42$) with the best or worst response, in terms of HFC reduction, to the intervention, and analyzed gut microbiota by the metagenomics data (Fig. 1c).

The microbiota data and clinical parameters were used to (1) characterize the change of gut microbiota composition in response to interventions; (2) determine associations of the microbiome with the level of physical fitness, fat mass, serum biomarkers and short-chain fatty acids (SCFAs); (3) analyze the metabolic function shift of the microbiome in response to interventions; (4) discover the co-occurrence network features of the microbiome ecosystem before and after intervention; and (5) predict personalized response to intervention based on the baseline gut microbiota composition and network.

**Changes of structure and function of gut microbiota induced by exercise and/or dietary intervention of participants with comorbidity of NAFLD and prediabetes.** We first performed 16 S rRNA gene sequencing of fecal samples collected before and after interventions and obtained a total of 5421 amplicon sequence variants (ASVs). We found that the alpha diversity (Shannon index) of gut microbiota was significantly decreased in the NI group ($p = 0.045$) with large individual variance by contrast to the intervention groups, which maintained their diversity after the intervention (Fig. 2a, NI vs. AED $p = 0.011$, NI vs. AEx $p = 0.007$ and NI vs. Diet $p = 0.025$, respectively, analysis of variance with repeated measures adjusted for change of body weight, baseline value and intervention duration). The changes in microbial diversity may reflect the natural progression of NAFLD in the NI group. Moreover, the principal coordinates analysis (PCoA) based on weighted UniFrac distances shows the

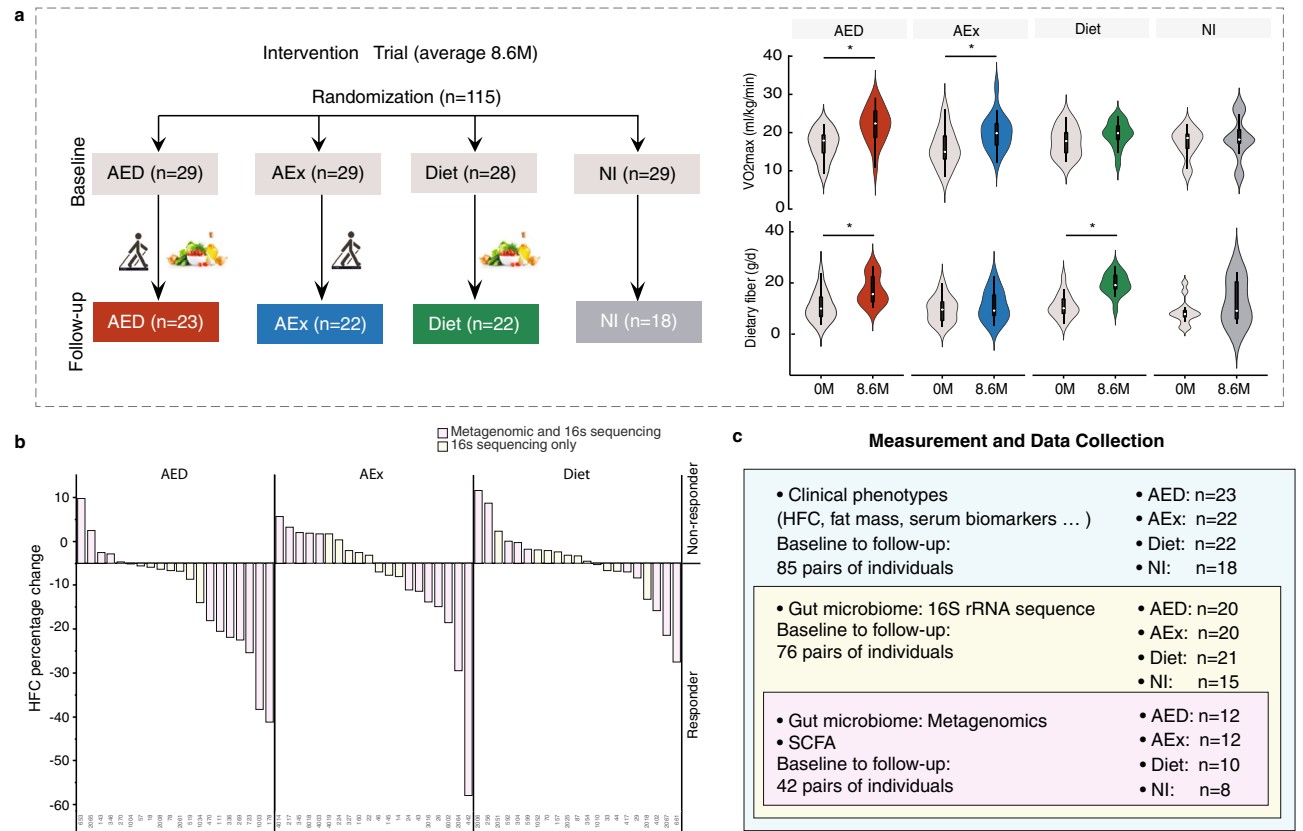

**Fig. 1 Summary of clinical study. a** Study design. The measurements were performed before and after intervention, and comparison of dietary fiber intakes and estimated physical fitness indicated by VO2max before and after intervention are given. The white point, box range, line range and density plot width represent median, interquartile range, 95% confident interval and frequence, respectively. AEx: aerobic exercise; AED: AEx+Diet; NI: no intervention. Comparisons within groups were performed using two-tailed paired t-test. *: Significance of $p < 0.05$. **b** Responders and low/non-responders according to change of HFC (the cutoff value was -5%) after intervention. AED: $n = 20$, AEx: $n = 20$, Diet: $n = 21$. **c** Data collection. Body fat mass was measured by DXA (dual X-ray densitometry); hepatic fat content (HFC) was determined by [1]HMRS (proton magnetic resonance spectroscopy); fecal samples were collected for microbiome assessments with both 16 s rRNA and metagenome assessments; and blood samples were used for clinical biomarkers assessments.

significant difference between NI and the other groups after the interventions ($p < 0.05$, PERMANOVA, Fig. 2b). These results indicate that microbial diversity deteriorates with the increased HFC, while exercise and dieting may help to maintain the diversity of the gut microbiota. To avoid only included the relative abundance data in the analysis which may introduce the bias, we then measured the absolute abundance of microbiome by quantitative real-time PCR (qPCR) targeting the 16 S rRNA gene. We found that there were no significant differences between baseline and after intervention as well as among the groups (Fig. S2a). Moreover, the total bacterial content of each subject in the same group at two time points also did not differ significantly (Fig. S2b–e).

Furthermore, we performed Linear discriminant analysis Effect Size (LEfSe) analysis[22] of ASVs that appeared in more than 20% of the samples. The ASVs showed significant differences between the intervention groups and the NI group after intervention (adjusted $p < 0.05$, log2 fold change >2) (Fig. 2c). Compared with the NI group, we found that 15 ASVs were enriched and 5 ASVs decreased in the AED group; and 13 ASVs were enriched in the AEx group, while 9 ASVs were enriched and 6 ASVs were decreased in the Diet group after the intervention. Among these ASVs, ASV2077 (AED: $p = 0.018$, AEx $p < 0.01$, Diet: $p = 0.032$) and ASV2513 (AED: p = 0.012, AEx: $p < 0.01$, Diet: $p = 0.013$) belonged to *Bacteroides*, and ASV3942 (AED: $p = 0.030$, AEx: $p = 0.024$, Diet: $p < 0.01$), which belongs to *Ruminococcus*,

increased in all intervention groups. In addition, ASV5361, which belongs to *Lachnospiraceae*, increased in both AEx ($p = 0.027$) and AED ($p = 0.037$) groups. ASV2440, belonging to *Bacteroides*, increased in both Diet ($p < 0.01$) and AED ($p < 0.01$) groups. Noticeably, some ASVs from the same family or genus showed different behaviors. For example, ASV 4432 in *Lachnospiraceae* was enriched in the AED ($p < 0.01$) group but decreased after the diet ($p = 0.023$) intervention compared with the NI group. To investigate whether changes of microbiome at the ASV level were similar to the changes at the genus level, we further performed a LEfSe analysis of the same parameters on the genus level. We found that those observed changed ASVs, if belonging to the same genus, indeed, they do have a similar trend at the genus level (Fig. S3).

We next performed partial Spearman correlation analysis to assess change of ASVs with clinical biomarkers after adjustment for body weight and fat mass ($n = 46$, Fig. 2d). We found that three altered ASVs were significantly negatively associated with the reduction of HFC [ASV2468, belonging to *Bacteroides* ($r = -0.31$, $p = 0.040$), ASV3307, belonging to *Ruminococcaceae* ($r = -0.32$, $p = 0.030$), and ASV4538, belonging to *Lachnospira* ($r = -0.37$, $p = 0.012$)]. In addition, ASV478, which belongs to *Phascolarctobacterium* ($r = -0.32$, $p = 0.033$); ASV1715, which belongs to *Alistipes* ($r = -0.35$, $p = 0.022$); and ASV 5195 and ASV5305, which belong to *Lachnoclostridium* ($r = -0.30$, $p = 0.045$, $r = -0.34$, $p = 0.023$, respectively), were negatively correlated with HbA1c. Only ASV776,

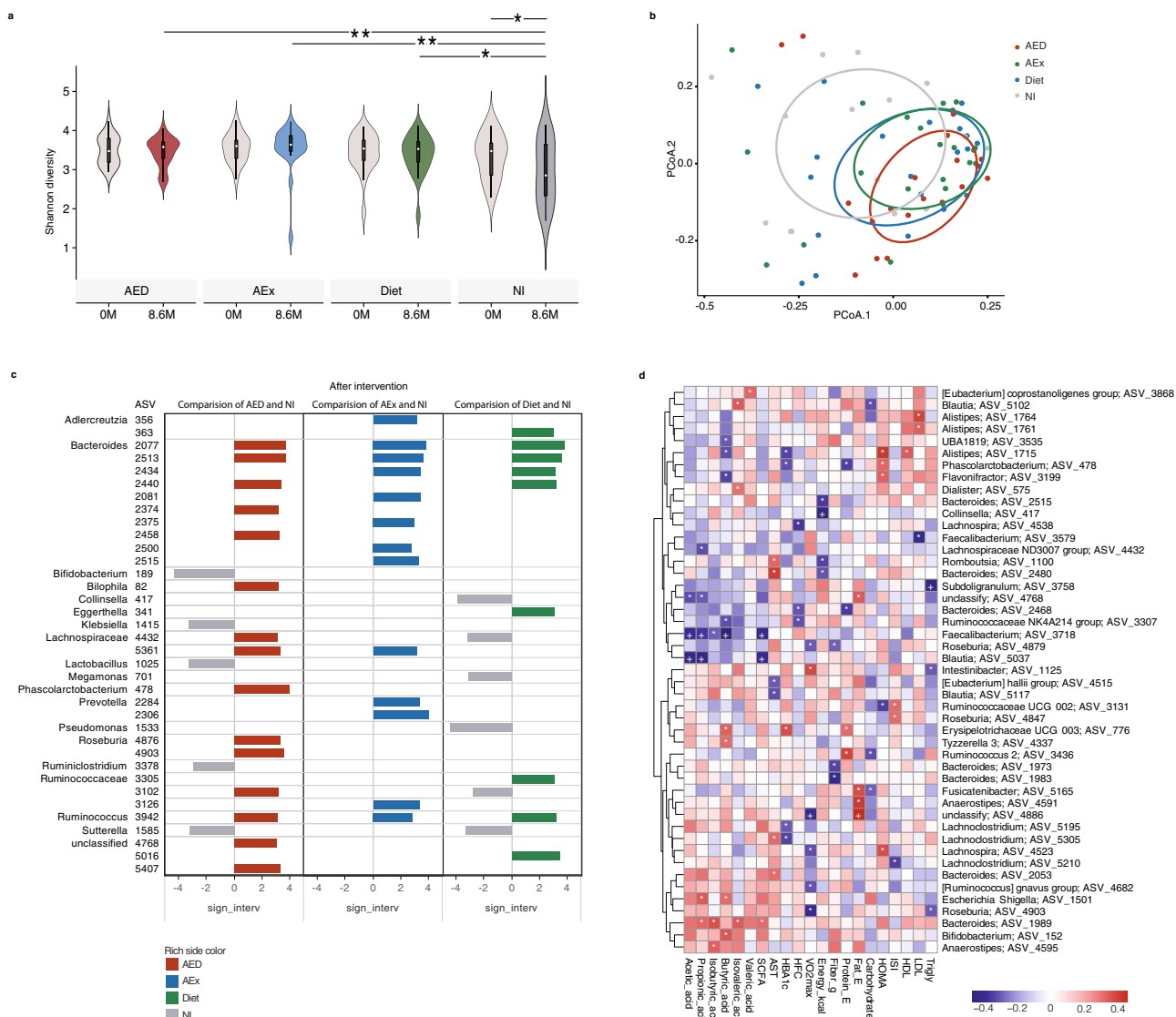

**Fig. 2 Changes in the structure of gut microbiota before and after intervention and relationship of microbial abundance with clinical parameters.**
**a** Alpha diversity (Shannon index) of gut microbiota was significantly decreased in the NI group by contrast to the intervention groups, which maintained their diversity after the intervention. The white point, box range, line range and density plot width represent median, interquartile range, 95% confident interval and frequence, respectively. ANCOVA for repeated measures (2 factor interactions: group × time) and controlled for change of body weight, baseline value and intervention duration followed by Sidak correction for multiple comparison between the groups. Contrast results (K Matrix) were used to localize the group differences: *$p < 0.05$, by contrast to the NI group. AED 8.6 m vs NI 8.6 m, $p = 0.011$; AEx 8.6 m vs NI 8.6 m, $p = 0.007$; Diet 8.6 m vs NI 8.6 m, $p = 0.025$; NI 0 m vs NI 8.6 m, $p = 0.031$. AED: $n = 20$, AEx: $n = 20$, Diet: $n = 21$ and NI: $n = 15$. **b** Weighted Unifrac distance of samples was calculated from the QIIME 2 ASV level. Ellipsoids represent a 50% confidence interval surrounding each group. **c** Differential ASVs between the AED/AEx/ Diet groups and NI group (LEfSe analysis, adjusted $p < 0.05$, log2 fold change > 2). Genus annotations on the left column covers multiple right ASVs. AED: $n = 20$, AEx: $n = 20$, Diet: $n = 21$ and NI: $n = 15$. **d** Heatmap of the Spearman's correlation coefficients between change of ASVs as assessed by 16 S and clinical parameters independent of body weight and fat mass. Statistically significant coefficients are marked by * and +, which means $p < 0.05$ and FDR < 0.1 respectively. Only ASVs with significant correlations (at least one based on p value) are shown.

belonging to *Erysipelotrichaceae* ($r = 0.36$, $p = 0.016$), was positively associated with HBA1c. Although a previous study[23] showed the gut microbiota may benefit humans via SCFAs production from carbohydrate fermentation, fecal SCFAs did not change significantly from baseline to follow-up within groups in this experiment. However, we found that some alterations in gut microbes correlated with changes in SCFAs (Fig. 2d). The level of butyric acid significantly correlated with 9 ASVs. Notably, ASV3718 in the genus *Faecalibacterium* was significantly negatively correlated with 4 out of all 6 SCFAs, while ASV1989 in the genus *Bacteroide* was positively correlated with 4 SCFAs (Fig. 2d).

To assess microbial functions, we also applied shotgun sequencing in a subset of samples (Fig. 1c) and annotated the data to the KEGG pathway (Fig. 3). By using LEfSe analysis ($p < 0.05$, LDA score > 2), we found that in total, 64 specific pathways were significantly different between the AED/AEx/Diet groups and NI group after interventions, most of which fall into the 'carbohydrate metabolism', 'energy metabolism', 'glycan biosynthesis and metabolism', 'lipid metabolism', 'amino acid metabolism' and 'metabolism of cofactors and vitamins' function pathways. The differences in functional pathways of the three intervention groups relative to the NI group were generally

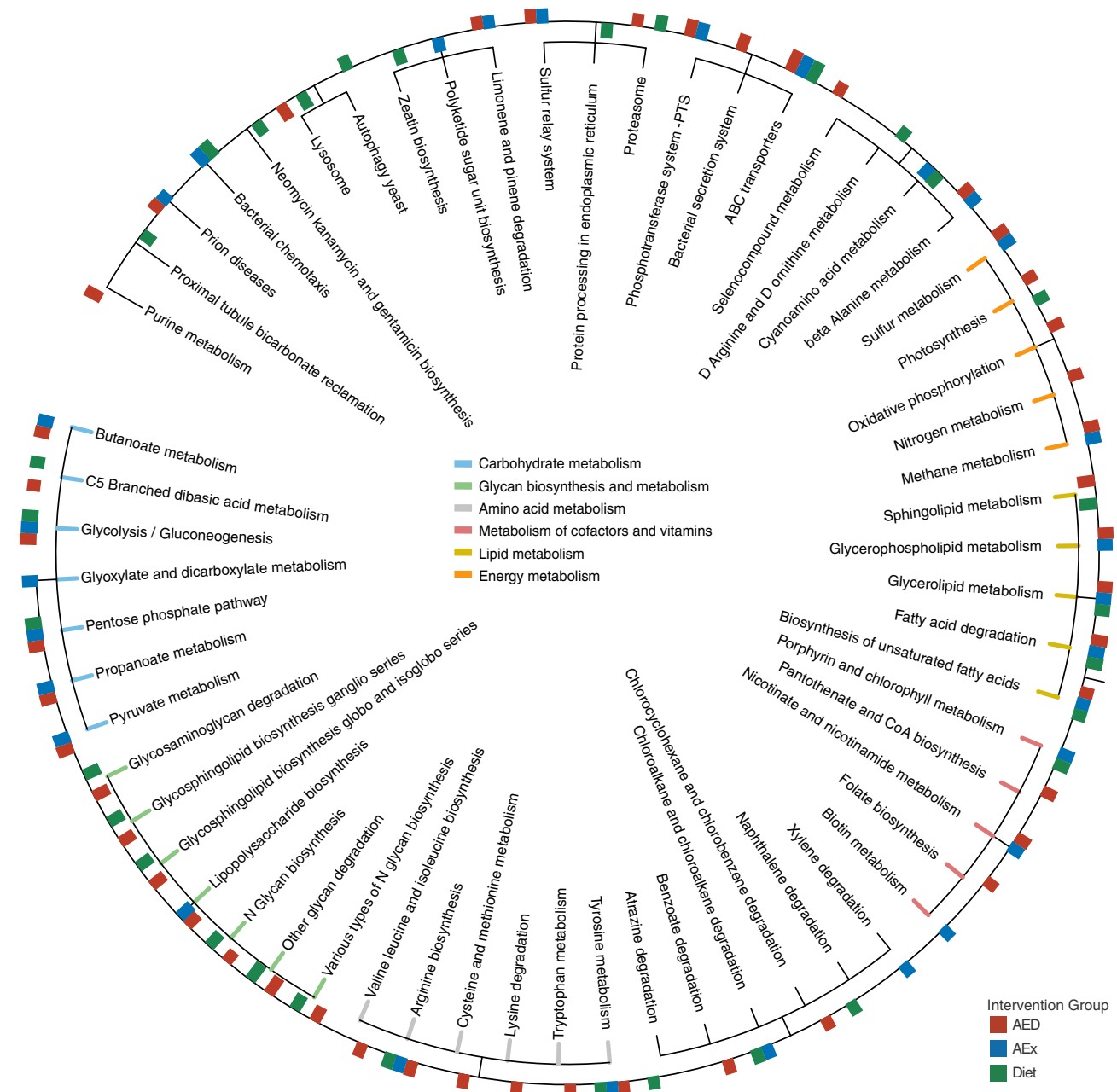

**Fig. 3 Identified pathways by KEGG between intervention groups and NI group after intervention.** Only significant pathways with a linear discriminate analysis (LDA) score > 2.0 and $p < 0.05$ are shown. Six colors indicate six primary function pathways. The bars inside the circle indicate that 3 intervention groups had higher values than the NI group, and the bars outside the circle indicate that the intervention groups had lower values than the NI group. The bar length represents LDA score.

similar, however, the AED group had more differential pathways than other groups. Notably, within 'lipid metabolism', 'Sphingolipid metabolism' was more vigorously different from that of other pathways (Fig. 3). In addition, most functional pathways were more abundant in the NI group; only 'glycan biosynthesis and metabolism' was more abundant in the intervention groups. Taken together, the above findings indicate that exercise or/and diet intervention(s) distinctly alter the abundance and function of microbiome, which are associated with changes in HFC and SCFAs.

**Intervention induced change in gut microbiota co-occurrence network**. It has been shown that bacterial species in the human gut may survive, adapt, and decline as interdependent functional groups (guilds) responding to environmental perturbations[24,25]. To identify bacteria in the gut ecosystem that responded as functional groups to interventions, we adopted "co-abundance groups (CAGs)" to analyze the community structure in the microbial ecosystem[24]. We used the SparCC algorithm to calculate the correlation coefficients among 279 ASVs shared by at least 20% of the samples from all the groups and time points[26]. These 279 ASVs were then clustered into 35 CAGs (Table S2) and their abundance difference between groups were showed in Fig. S4a.

Network analysis can disentangle microbial co-occurrence and provide comprehensive insight into the microbial assembly patterns and community structures[27]. Using Spearman correlation analysis, we constructed a co-occurrence network (Fig. 4a–d) to illustrate the potential interactions among the 35 CAGs in each group with

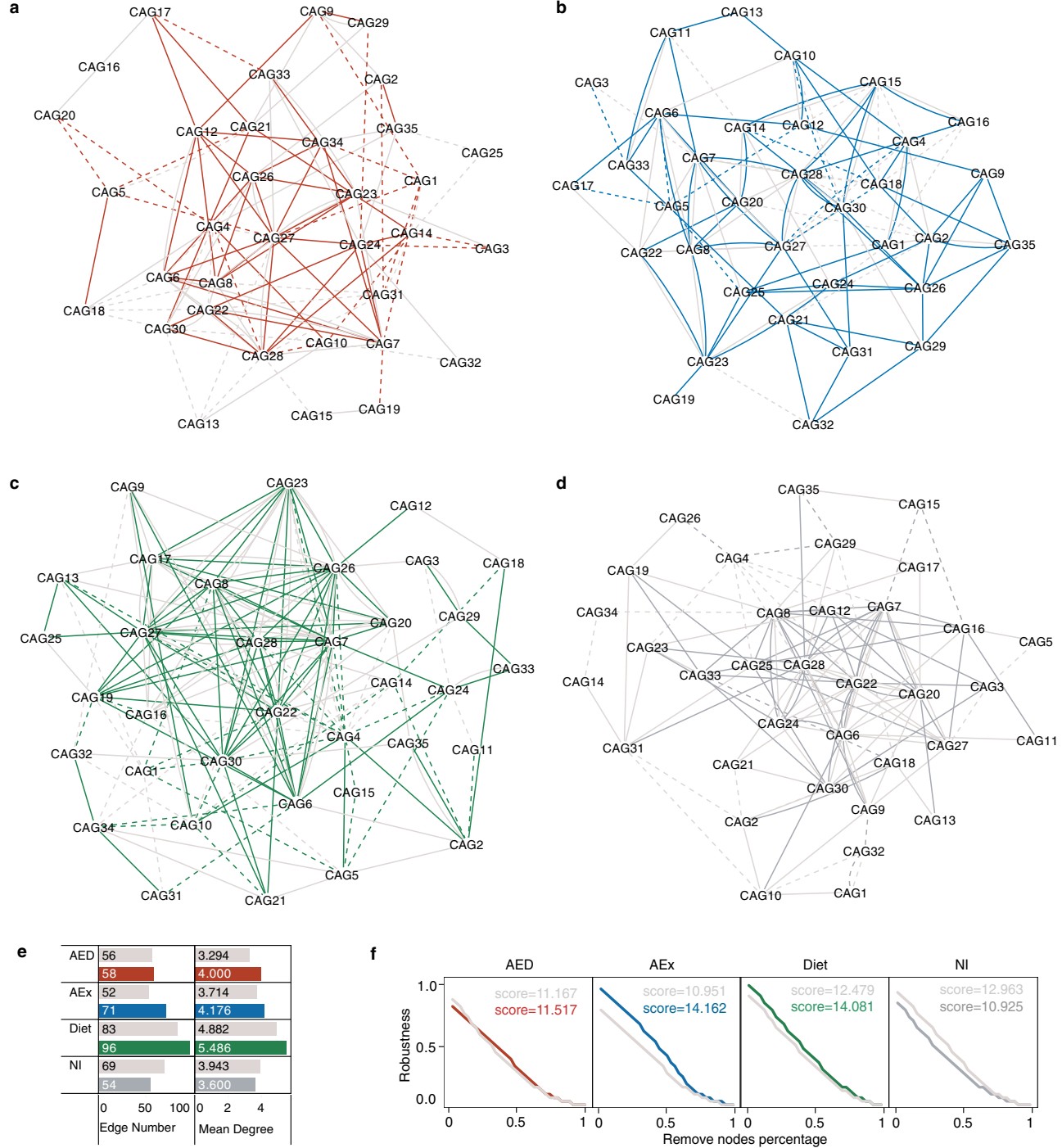

**Fig. 4 Co-occurrence networks before and after intervention in paired participants. a–d** Co-occurrence networks of CAGs (co-abundance groups) before and after intervention in different groups (**a**: AED: $n = 20$; **b**: AEx: $n = 20$; **c** Diet : $n = 21$; **d**: NI: $n = 15$). The nodes represent CAGs and the edges represent statistically significant differences ($p < 0.05$) via Spearman correlation between CAGs without adjustment. The style of the edge is set to a dashed line and a solid line, which represent negative and positive correlation, respectively. Gray edges indicate correlation before intervention; colored edges indicate correlation after intervention. **e** Network connection properties. **f** Network robustness test.

baseline and follow-up samples. Each node in the network represents a CAG, and each edge represents a significant correlation ($p < 0.05$) between two CAGs. We found that the number of edges and average degrees that signify the connectivity of CAGs in the gut microbiota network were higher after intervention than at baseline in the AED, AEx and Diet groups, but not in the NI group (Fig. 4e). In addition, the network robustness[28] was significantly decreased in the NI group but maintained or increased in all intervention groups

(Fig. 4f). The greatest increase in robustness score was in the AEx group (+29.3%), followed by the Diet (+12.8%) and AED ( + 3.1%) groups.

Because the complexity caused by the nonlinear dynamics of the nodes is related to the network architecture, we assessed the distribution of the network degree to judge the topological features of the microbial network (Fig. S4b). We found that most of the connections of the co-occurrence network were

concentrated in a few nodes, which is an important features of a scale-free network. This kind of network with a power-law distribution comprise highly connected nodes, which are defined as hubs[29]. The hubs in a microbial network have been proposed as keystone taxa, as their removal has been computationally shown to cause a drastic shift in the composition and functioning of a microbiome[20]. In our data, the nodes (CAG6, CAG7, CAG8, and CAG28) had more than 4% of the total number of connections in each network. Therefore, these CAGs can be regarded as hubs and maybe the keystone taxa of the gut microbial community of each group (Fig. S4c). Interestingly, the abundance of these four CAGs increased significantly after intervention in the AED group compared with the NI group, but did not change significantly in the AEx and Diet groups (Fig. S4a). The most abundant ASVs among these four CAGs belong to *Bacteroides*, *Ruminococcaceae*, *Alistipes* and *Subdoligranulum*. Collectively, these findings suggest that interventions improve the stability of the microbial ecosystem by reconstructing the network among keystone taxa.

**Personalized gut microbial networks can predict the intervention efficiency of individual patients**. Since not all subjects responded to the intervention in a similar manner (Fig. 1b), we assessed whether these response variations were associated with the individual characteristics of the gut microbiota. First, we used differentially altered ASVs abundance in each intervention group (Fig. 2c) to classify the responders and low/non-responders, we found that only the baseline ASVs abundance in the Diet group showed the prediction power with a receiver operating characteristic (ROC) area under the curve (AUC) of 0.65 (specificity = 0.58, sensitivity = 0.67). No significant prediction power for the classification in the AED group (AUC = 0.53, specificity = 0.40, sensitivity = 0.67) and the AEx group (AUC = 0.52, specificity = 0.50, sensitivity = 0.60) was found. Second, since ASVs abundance could not describe the individual network characteristics, we developed a Single SparCC network method by combining a SparCC network[26] with a sample specific network (SSN)[30] (Fig. 5a). We used baseline metagenomics data to build a network for each patient (Fig. 5b) and found that responders in all three intervention groups tended to have more interactions between species than low/non-responders, but no statistical significance was observed (Fig. S5). Furthermore, in the linear regression analysis, the edge number in the AEx group significantly predicted the change of HFC (Fig. 5c). Then we performed an unsupervised classifier to differentiate the responders from the low/non-responders by each Single SparCC network edge number. We found that AUC was about 70% in different intervention groups (Fig. 5d), and the AUC of supervised Least absolute shrinkage and selection operator (LASSO) classifiers were higher in all three groups (Fig. 5e). For the purpose of comparison, we also tested the ability of age, body weight (WT) and body mass index (BMI) to distinguish the responsiveness of the interventions (Fig. S6). The result showed that these clinical parameters were not able to differentiate the responders from the low/non-responders (age: AUC = 0.437, p = 0.347; WT: AUC = 0.470, p = 0.655; BMI: AUC = 0.519, p = 0.771), except for BMI in the AED group.

Considering the small sample size of metagenomic data, we performed the same Single SparCC network analysis on the intervention groups' samples by using the 16 S rRNA gene sequencing data (Fig. S7). Consistent with the metagenomic results, the responders of the three intervention groups tended to have more network edges than low/non-responders, but none of them were statistically significant (Fig. S8a). In regression analysis, we found that, in addition to the AEx group, the

baseline edge numbers in the AED group also showed significant correlation with the HFC change; however, this was not observed in the Diet group (Fig. S8b). For supervised LASSO classifiers, ROC showed that the edge number of ASV data used to predict the responder exceeded 70% AUC in all three intervention groups (Fig. S8c), which was higher than the metagenomics data. However, the effect of unsupervised classifiers deteriorated and was almost ineffective in the Diet group (Fig. S1d). These results indicated that our Single SparCC networks could be used to construct personalized gut microbial networks for each individual sample and thus differentiate the responders from the low/non-responders, particularly for the exercise intervention. Taken together, these analyses demonstrate that the individual baseline gut microbial network can predict the response to exercise intervention.

## Discussion

In the current study, in patients with NAFLD and pre-diabetes, we identified the characteristics of the gut microbiota responding to an 8.6-month aerobic exercise and/or low-carbohydrate dietary intervention. We showed that after combined exercise and diet intervention, changes are prominent in gut microbial composition in which keystone taxa became divergent but their connections remained stable. By contrast, with exercise or diet intervention alone, effects were significant regarding network connectivity and robustness between taxa. Moreover, the personalized microbial network was able to predict the intervention efficiency of individual patients.

Members from *Ruminococcus* have been reported to produce SCFAs from complex carbohydrates[31,32]. A recent study[33] showed that *Ruminococcaceae* was negatively correlated with the fibrosis severity. In agreement with this, members in *Ruminococcaceae* was found negatively correlation with HFC in our study. In addition, some members of *Bacteroides* contribute to the release of energy from dietary fiber and starch, and they are likely to be a major source of propionate[34]. Accordingly, we found that members in *Bacteroides* was positively correlated with propionic acid, isobutyric acid and isovaleric acid. A previous study showed that compared to western countries, the abundance of *Bacteroides* in NAFLD is much lower in Chinese individuals[35]. However, our result suggested an important energy-extracting role of members in *Bacteroides*, although their abundance is relatively low in Chinese population. Of note, there are currently conflicting results about the shifts of microbiota at the taxon level in NAFLD-related studies. For example, some studies showed that *Bacteroides* abundance was higher in NASH (nonalcoholic steatohepatitis, a stage of NAFLD progression) than in non-NASH[36]. Another article showed the phylum *Bacteroidetes* (44.63%) tended to be more abundant in healthy subjects than in NAFLD[35]. Our results showed that members in *Bacteroides* abundance was increased after both diet and exercise intervention, and this increase was correlated with decreased HFC. However, previous studies showed higher fiber intake was marginally associated with lower abundance of *Bacteroides uniformis*[37], and *Bacteroides* was increased in participants with obesity but decreased in lean participants after exercise intervention[38]. These inconclusive results may be ascribed to differences in ethnicity, living environment and lifestyle among the cohorts. Thus, the mechanism of action of the key microbiota needs to be investigated in future studies.

The long-term adaptation after intervention may account for maintenance of gut microbiota alpha diversity in all intervention groups in contrast to the NI group in which alpha diversity decreased. However, the connectivity and robustness of the microbiota co-occurrence networks improved in three

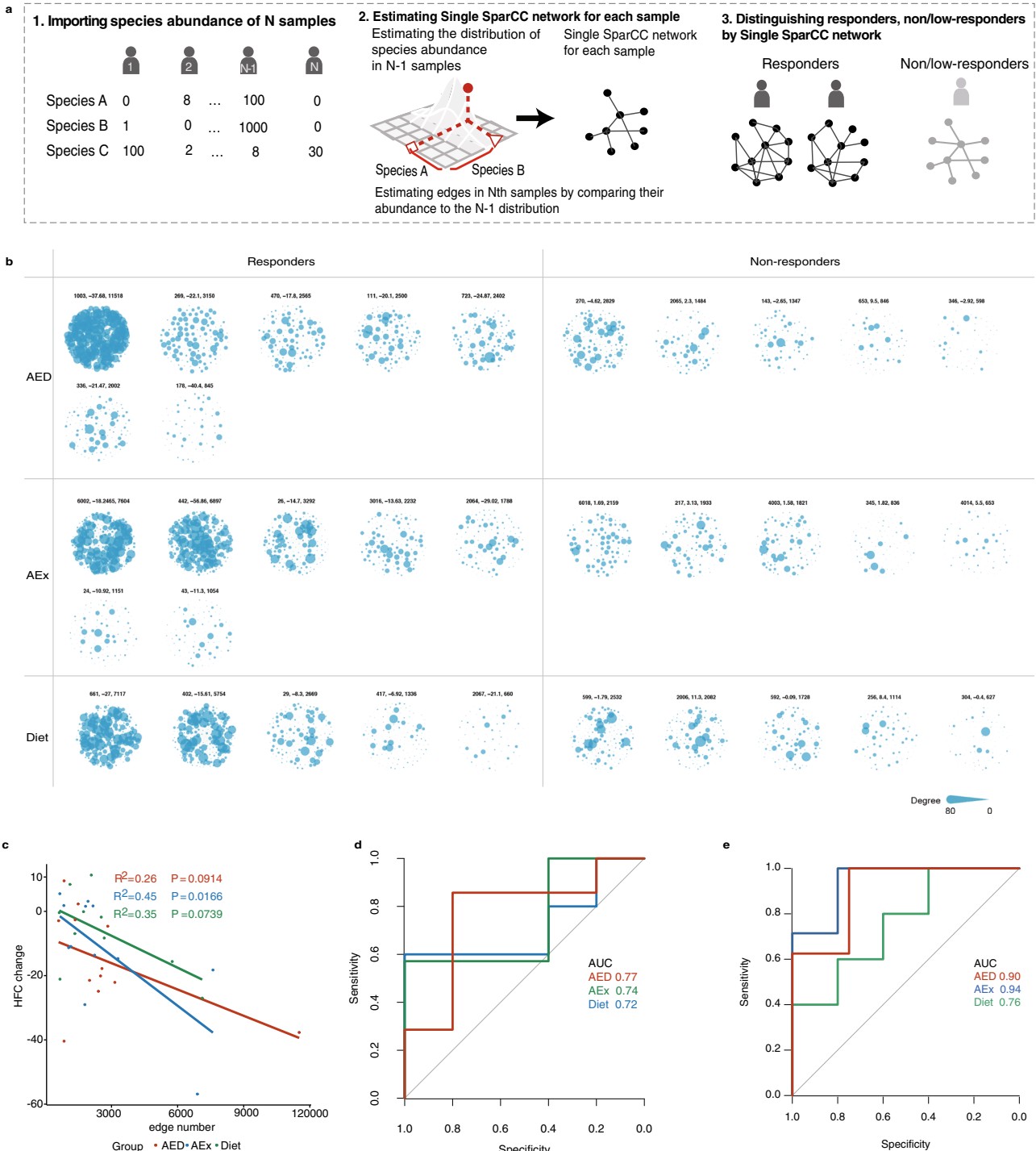

**Fig. 5 Intervention efficiency predicted by individual baseline gut microbial networks on metagenomics species data. a** Prediction of responders using our single SparCC network method. **b** The gut microbial network for each individual before intervention based on metagenomics species data by the Single SparCC network method. Three numbers under every network represent subject ID, HFC change and edge numbers. **c** Linear regressions of edge number (Single SparCC network) with the change of HFC after intervention. AED: $n = 12$, AEx: $n = 12$, Diet: $n = 10$. **d** ROC curve of edge number for differential responders from low/non-responders within each intervention group by unsupervised classification. AED: $n = 12$, AEx: $n = 12$, Diet: $n = 10$. **e** ROC curve by supervised Lasso model performance. AED: $n = 12$, AEx: $n = 12$, Diet: $n = 10$.

intervention groups. Various ecological microbiome studies[39,40] have shown that healthy people have a higher average network degree and hence greater connectivity of their gut microbiota network[41]. A poorly developed microbial network usually has lower functionality due to fewer taxa present that can support their function in ecosystems[39]. In combination with these findings, our results indicate that microbiota as an ecosystem were more healthy and stable after intervention. Interestingly, when

ASVs abundance is used to predict the change of HFC, it has the best predictive power in the diet group, but is not predictive in the exercise group. However, when assessed the robustness of co-occurrence networks, the exercise intervention showed the highest increment of robustness score. A possible explanation is that diet intervention directly affects the structure and composition of gut microbiota through changes in nutrients, while exercise intervention influences the interaction between bacteria through the regulation of host metabolism. In addition, this study confirmed that co-occurrence networks of microbiota are scale-free networks, and further defined four keystone taxa, which have an prominently high degree in each network. A recent study found that keystone taxa that are highly connected in the microbiome can explain microbiome compositional turnover better than all taxa combined[42]. Therefore, we speculate that what exercise intervention improves not the gut microbiota diversity, but the stability of the interaction network of the gut microbiota ecosystem, thereby improving the metabolism of the host. The keystone taxa determined by the microbiota population network topology may be a new perspective beyond finding key bacteria by abundance and diversity comparison.

No disease-specific drug/medical therapy for NAFLD is available due to its complexly pathogenetic mechanism, and management of lifestyle (such as exercise and diet) in patients is the only effcient invervention in clinical practices. However, studies have shown that not all participants respond to invervents of exercise or diet in a similar way[16], which may be attributable to the individual-variation and response of gut microbiota in patients. In some studies, 50% of the study participants show no change in their gut microbiota composition after an exercise intervention[17]. Then prediction of inprovment of clinical phenotypes in different patiens based on their gut microbiota would be important to increase effcieny of NAFLD treatment. In the previous studies, they ascribed to use the group-level changes of bacteria abundance to construct the prediction models, which disregards the fact that the menbers of gut microbiota show ecological interactions such as competition or cooperation with each other. Our Single SparCC method evaluated not only microbiome composition and abundance, but also the interactions between species. Such a network can be used to quantify the system-wide features, e.g., robustness or stability of the system, and further predict intervention effects. Various studies[28,43] have adopted the concept of connectivity and robustness from network theory to investigate the rule of microbial community in diseases from an ecological perspective. Actually, there is increasing evidence indicating a link between undesirable health conditions with altered microbial assembly process and a more fragile microbiome network[40,41]. Our results indicated that the Single SparCC network of gut microbiota is a valid method to predict the responders from low/non-responders for exercise intervention and therefore opens a new avenue to assess the effect of exercise intervention on gut microbiota at an individual level. Interestingly, we found that when differentially altered ASVs abundance is used to classify the responders and low/non-responders, it has the best prediction power in the diet group but invalid in the exercise group. In terms of the AUC values, the Single SparCC network of gut microbiota was better differential features to responders from low/non-responders than age, body weight and BMI in our study. These results indicate that corresponding gut microbial characteristics can be used to predict responsiveness of a subject to specific intervention methods. This is of great clinical significance in improving effectiveness and efficiency of interventional therapy of NAFLD. Notably, the personalized prediction here requires a number of samples or other individuals measured as the reference samples, against which the network of each test sample can be projected.

Comparing with the conventional machine learning methods, although the reference samples can be viewed as the training samples, our method can further extract the second-order statistical information (e.g., correlation or association or network) of each individual approximately, thus providing system-wide features of each test sample. In general, our results are important in terms of how to assess the intervention effect on physiology and pathophysiology and to develop personalized lifestyle treatment for such patients through their individual gut microbiota network.

Our study provides novel information on how exercise and diet intervention affect the gut microbiome for patients with comorbidity conditions at both the population and individual levels. However, there are some limits in this study. Our sample size for metagenomics is relatively small but it is comparable with most of the studies[44]. The dietary and exercise intervention methods are often different from trial to trial, and even if the intervention methods are the same, the intervention contents and intensity are different which makes the comparison with other trials challenging. In addition, since the trial was not originally designed to predict response efficiency with personalized microbiota network, our predictions were not validated by independent samples. Nevertheless, this study provides a new perspective in the research of gut microbiota, that is, in addition to focusing on the differential strains themselves, the topological properties of the microbial ecosystem community network are also of great significance. Using the network features of gut microbiota to predict individual response to exercise and diet interventions may help to choose more precise and effective treatment for different NAFLD pateints.

## Methods

The study was approved by the Ethics Committee of Shanghai Institute of Nutrition (06.01.2013) and has been retrospectively registered in the International Standard Randomized Controlled Trial Number Register (ISRCTN 42622771). We acknowledge that one limitation of the study is that the trial was restrospectively registered. Two of the primary outcomes (Glucose tolerance test and Liver fat content) have been previously reported[21]. All participants signed informed consent forms before beginning the study. The study was conducted in accordance with the principles of the Declaration of Helsinki.

**Subjects and design.** This study was a randomized controlled trial. Detailed information was given in our previous publication[21]. The enrollment of participants was between January 6 and December 25, 2013. The eligibility criteria for intervention groups were as follows: men or women aged 50–65 years with impaired fasting glucose (IFG between 5.6 to 6.9 mmol/L) or impaired glucose tolerance (IGT between 7.8 to 11.0 mmol/L 2 hour after the intake of 75 g glucose), and diagnosed as NAFLD by [1]H MRS (liver fat > 5%) and by questionnaire that ongoing or recent alcohol consumption is <21 drinks on average per week in men and <14 drinks on average per week in women; no chronic cardiovascular, serious musculoskeletal or gastrointestinal problems and not on extreme diets; and for women, serum follicle-stimulating hormone level greater than 30 IU/L and last menstruation more than 6 months ago but within 10 years. The exclusion criteria included body mass index (BMI) > 38 kg/m$^2$; serious cardiovascular or musculoskeletal problems; diagnosed with Type 1 diabetes and T2D; and mental illness. In the initial study design, we estimated the sample size based on previous literature of liver fat and microbiota as primary outcome variable[21]. We estimated that 34 subjects in each group would have 85% power for mean comparison between the groups and therefore we targeted to have 50 subjects in each group. However, during the recruitment period, we found that only about 20% of the subjects met the inclusion criteria and due to the limited funding, we re-calculated the sample size based on our previous similar type of study[45,46]. Thus, for this report, sample size estimation for the primary outcome HFC with 29 individuals has 95% power to test against the null hypothesis that there is no change in any group. Further, when intervention groups compared with NI group having 17 subjects, the power for the HFC was 84% with < 0.05 two-sided significance level. After every subject reached minimal 6 months intervention, the study did not continue to follow-up all the subjects and consequently, the follow-up study was terminated. There were no sex distribution differences between the groups and the male/female ratio for each group was n = 6/23 for AEx, n = 6/22 for Diet, n = 7/22 for AED, and n = 7/22 for NI group, respectively.

**Interventions.** The duration of the intervention ranged from 6.5 months to 11.1 months and on average 8.6 months. There were no significant differences in

the intervention duration among the groups (AEx 8.75 months, Diet 8.6 months, AED 8.5 months and NI 8.6 months) as well as between responders (8.6 months) and non-responders (8,7months).

The exercise (AEx) group participated in a supervised progressive aerobic exercise training program (such as Nordic brisk walking + stretching and other group exercises). Exercise was performed 2–3 times a week, 30–60 min per session with 60–75% of the maximum oxygen uptake (estimated from fitness test). Each exercise session included a 5 min warm-up and 5 min cool-down period. The diet (Diet) group received a daily prepared meal (lunch), which accounted for 30–40% of the total daily energy intake on the basis of each individual's dietary intakes and body weight. The meal included 37–40% carbohydrate with 9–13 g as fiber, 35–37% fat (SAFA 10%, MUFA 15-20%, PUFA 19 10%) and 25–27% protein plus 5 g of soluble fiber (dietary water-soluble fiber). The lunch box was then delivered to the study district community office where the study subjects were gathered. If the participants did not have time to come to pick up the meal, it was delivered to their home and the participants ate it for dinner. The other two meals (breakfast and dinner) were cooked by subjects themselves following the nutritionist advice. In addition, the Diet group was advised to maintain their current level of physical activity during the intervention. The Exercise plus Diet (AED) group performed the same exercise program and followed the same diet as described above for the AEx and Diet groups. The no intervention (NI) group was advised to maintain their current level of physical activity and eating habits during the intervention.

**Patient and public involvement**. The patients were engaged via advertisements distributed in 7 health clinical service centers in the Shanghai Yangpu district. In addition, media and social media were used to inform about the study. Patients were not involved in the design of the study. The study protocol was described to study participants on the first study visit.

**Fecal DNA extraction**. Faecal samples were collected at the baseline and after the intervention. Microbial genome DNA was extracted from fecal samples using the bead-beating method (alterations in the gut microbiome and metabolism with coronary artery disease severity). Briefly, we used an InviMag® Stool DNA kit (Invitek, Berlin, Germany) with agitation in a mini-bead beater (Biospec Products, Bartlesville, OK, USA) as follows: approximately 0.2 g of thawed feces were added to a 2 mL screw-cap tube containing 1 mL of lysis buffer P of the kit and 0.3 g Zirconia beads (0.1 mm, Biospec Products, Bartlesville, OK, USA). After sufficient homogenization by vortex for approximately 5 min, bead beating was performed for 1 min at maximum speed. DNA was extracted by following the manufacturer's instructions for bacterial DNA extraction involving proteinase K treatment and subsequent purification using the KingFisher device (Invitek, Berlin, Germany).

**qPCR of total fecal bacteria**. A plasmid of the 16 S full-length positive Ruminococcus strain ($n = 1010$ copies/µl) was diluted according to different gradients successively to 109, 108, 107, 106, 105, 104, 103, and 102 copies/µl. qPCR was performed in a 20-µl reaction system containing template (20 ng), primer Uni331F (5′-TCCTACGGGAGGCAGCAGT-3′), primer Uni797R(5′-GGACTACCAGGGT A TCTAATCCTGTT-3′), and supermix (Bio-Rad) on a qTOWER3G touch system, with 2 replicates for standard and sample DNA. The PCR conditions were 95 °C for 5 min, followed by 40 cycles of 95 °C for 20 s, 60 °C for 60 s, and 80 °C for 5 s, and Melting curve 60 °C to 95 °C. A standard curve was determined though a linear fifit of the copy number and CT value of the plasmid in different gradients. The copy number of sample DNA was calculated through a standard curve.

**Construction of amplicon library of V3-V4 region of 16 S rRNA and sequencing**. The extracted DNA from each sample was used as the template to amplify the V3-V4 region of 16 s rRNA genes with bar-coded primers, forward primers-338F (5′- sampleIDtag- ACTCCTACGGGAGGCAGCA-3′) and reverse primers-806R (5′- sampleIDtag-GGACTACHVGGGTWTCTAAT). PCR reactions were run in a thermocycler PCR system (Applied Biosystems, USA) using the following programme: 5 min of denaturation at 96 °C followed by 25 cycles of for 30 sec at 96 °C (denaturation), 30 sec at 50 °C (annealing) and 30 sec at 72 °C (elongation), with a final extension at 72 °C for 5 min. PCR product was excised from a 1.5% agarose gel and purified by QIAquick Gel Extraction Kit (QIAGEN, cat# 28706). Purified PCR products from the fecal samples were combined at equal concentrations and used to construct a metagenomic library using an Illumina TruSeq sample preparation kit (Illumina, San Diego, CA, USA) according to the manufacturer's protocol. Barcoded V3-V4 amplicons were sequenced using a pair-end method via the Illumina Miseq (Illumina, San Diego, CA, USA) sequencing platform with a 6-cycle index read.

**16 S rRNA Sequence analysis**. Quality control of the raw data was performed as follows: (1) the sequence should have a perfect match to the barcode in at least one end; (2) the sequence should have a BLAST match to at least one end of the 16 S rRNA gene V3-V4 region primers; (3) the length of the trimmed sequence (without barcodes/primers) should be between 400 nt and 500 nt; and (4) there should be no more than two undetermined bases. All the retained sequences were processed using the QIIME 2 (Quantitative Insights into Microbial Ecology) package. The sequences were aligned using the PyNAST aligner with the Greengenes core set

(Released 13.8) and then classified into ASVs at a threshold of 97% sequence identity using UCLUST. The representative sequence for each ASV was selected using default parameters and was imported into the latest Greengenes ARB database to construct a phylogenetic tree.

Raw sequencing data were analyzed by QIIME 2. In the process of running the DADA2 pipeline, based on the quality profile of the data, forward and reverse reads were trimmed accordingly to ensure that the median quality score for each position was above 32. The taxonomy of all ASVs were annotated by the SILVA (v132) reference database[47]. All samples were rarefed to 10,000 per sample for downstream analysis.

**Metagenomics analysis**. Forty-two paired samples were sequenced at the Novogene Bioinformatics Institute (Novogene, Beijing, China). After construction of 350 bp insert libraries, metagenomic samples were pair-end sequenced on the HiSeq X Ten System. Cluster generation, template hybridization, isothermal amplification, linearization, and blocking denaturing and hybridization of the sequencing primers were performed according to the workflow specified by the service provider. The pipeline was consistent with previous studies[48,49]. We removed contamination of host DNA and assembled the metagenome by using Bowtie (Default parameter Settings:—end-to-end,—sensitive, -i 200, -x 400) and SOAP denovo (K-mers = 55)[50].

**Short chain fatty acid measurement**. Slice of human feces were homogenized with 1.2 mL of phosphate buffer solution in weighed 2 mL centrifuge tubes (marker as W1). The mixture was vibrated completely and centrifuged twice at 4 °C at $16000 \times g$ for 15 min. The supernatants were filtered and sterilized through a 0.22 µm nylon filter (EMD Millipore). Feces residue and tube were dried in 80 °C drying apparatus overnight and weighed (marked as W2). 200 µL of the supernatant was acidized with 100 µL of 50% (v/v) sulfuric acid. After vortexing and standing for 2 min, the organic phase was extracted by adding 400 µL of diethyl ether. The extracted solution was measured by gas chromatography (GC) on an Agilent 6890 (Agilent Technologies, CA, USA). The concentrations of 6 short chain fatty acids (acetic acid, propionic acid, butyric acid, isobutyric acid, valeric acid and isovaleric acid) were calculated by each peak area and feces dry weight (W2-W1).

**Co-abundance gene groups (CAGs)**. The ASVs shared by at least 20% among all samples were considered key ASVs. The correlation among 279 key ASVs was calculated by the SparCC algorithm[51] with a bootstrap procedure repeated 100 times, and then correlation matrices were computed from the resampled data matrices. Once the bootstrapped correlation scores have been computed, only ASVs with correlation scores greater than 0.4 were classified into CAGs. The p value at the desired cut-off < 0.05.

**Network fragility**. Given the N node (CAG) number in a network, the N nodes were removed one by one in a random order. The fragility statistics (y axis) were the ratio of the node number in the largest connected component to the node number N, and were calculated when each node was removed, while the percentage of removed nodes was taken as the x axis. The robustness value R for a network was related to a function $\sigma$, which was dependent on the decreasing order of fragility statistics as nodes were removed[28], i.e.,

$$R = \frac{1}{N} \sum_{i=1}^{N} \sigma(i/N)$$

where $i$ represents the ith CAG. To ensure the results were stable, we repeatly remove the nodes in each network for 1000 times. The median fragility statistics of the 1000 curves were taken as the coordinates for the final curve to plot the robustness curve.

**Single SparCC network analysis**. The traditional correlation coefficient mainly exploits the general information in a group of samples. Rather than focus on a 'mean' value within a group, our Single SparCC network is proposed to investigate the characteristics of each sample. Gut microbiota 16 S sequencing and metagenomics data suffered from spurious correlation. SparCC is the most popular method to eliminate spurious correlation[26]. SparCC can detect the association between species in a set of sample size > 1, but it is invalid for an individual sample. Our approach enabled us to explore the personalized characteristics of NAFLD patients.

The correlation of species $i$ and $j$ can be inferred from their ratio. Even though high-throughput sequencing only produce relative quantification of microbe, the ratio of those relative value is the same with real abundance. Define $t_{ij}$ to describe variance of log transformed relative value of component $x_i$ and $x_j$:

$$t_{ij} \equiv \mathrm{Var}\left[\log \frac{\omega_i}{\omega_j}\right] = \mathrm{Var}\left[\log \frac{x_i}{x_j}\right], \qquad (1)$$

where $x_i$ and $x_j$ are the relative abundances of microbe $x$ and $y$, while $\omega_i$ and $\omega_j$ are the absolute abundances of those two microbes.

In SparCC method, $t_{ij}$ was related to Pearson correlation coefficient $\rho_{ij}$ by equation:

$$t_{ij} = \omega_i^2 + \omega_j^2 - 2\rho_{ij}\omega_i\omega_j, \qquad (2)$$

where $\omega_i^2$ and $\omega_j^2$ are the variances of the log-transformed real abundance of species $i$ and $j$. Because of the sparse nature of microbial association network, we can assume that the average of all $\rho_{ij}$ correlation is small enough to ignore, which allow us to solve for $\omega_i^2$ and $\omega_j^2$. Then put in the values of $\omega_i^2$ and $\omega_j^2$ back to Eq. (2), and end up with the approximated correlation $\rho_{ij}$. Suppose the sample size is $S$ and the number of nodes is $D$. Letting $a = 2D\text{-}3$, $b = \frac{1}{2(D-1)(D-2)}$, we achieve an analytical solution for $\rho_{ij}$:

$$
\rho_{ij} = \frac{1}{2s\omega_i\omega_j}\sum_{k=1}^{S}\left(b(a+1)\sum_{j=1}^{D}\left(\log\frac{x_i^{(k)}}{x_j^{(k)}} - \mu_{\log x_{ij}}\right)^2 \right.
$$
$$
\left. - 2b\sum_{j=1}^{D}\sum_{i=1}^{D}\left(\log\frac{x_i^{(k)}}{x_j^{(k)}} - \mu_{\log x_{ij}}\right)^2 + b(a+1)\sum_{i=1}^{D}\left(\log\frac{x_i^{(k)}}{x_j^{(k)}} - \mu_{\log x_{ij}}\right)^2 \right. \qquad (3)
$$
$$
\left. - \left(\log\frac{x_i^{(k)}}{x_j^{(k)}} - \mu_{\log x_{ij}}\right)^2\right)\right)
$$

where

$$
\omega_i^2 = b(a+1)*\frac{1}{S}\sum_{j=1}^{D}\sum_{k=1}^{S}\left(\log\frac{x_i^{(k)}}{x_j^{(k)}} - \mu_{\log x_{ij}}\right)^2
$$
$$
- b*\frac{1}{S}\sum_{i=1}^{D}\sum_{j=1}^{D}\sum_{k=1}^{S}\left(\log\frac{x_i^{(k)}}{x_j^{(k)}} - \mu_{\log x_{ij}}\right)^2. \qquad (4)
$$

$$
\mu_{\log x_{ij}} = \sum_{j=1}^{D}\sum_{k=1}^{S}\left(\overline{\log\frac{x_i^{(k)}}{x_j^{(k)}}}\right), \qquad (5)
$$

Since $\rho_{ij}$ is a form of k elements summation, we can break it down into a vector that depends on the correlation coefficient $\rho_{ij}$, and each element $\rho_{ij}^{(k)}$ is the sample specific correlation:

$$
\rho_{ij}^{(k)} = \frac{1}{2s\omega_i\omega_j}\left(b(a+1)\sum_{j=1}^{D}\left(\log\frac{x_i^{(k)}}{x_j^{(k)}} - \mu_{\log x_{ij}}\right)^2 - 2b\sum_{j=1}^{D}\sum_{i=1}^{D}\left(\log\frac{x_i^{(k)}}{x_j^{(k)}} - \mu_{\log x_{ij}}\right)^2\right)
$$
$$
\left(+ b(a+1)\sum_{i=1}^{D}\left(\log\frac{x_i^{(k)}}{x_j^{(k)}} - \mu_{\log x_{ij}}\right)^2 - \left(\log\frac{x_i^{(k)}}{x_j^{(k)}} - \mu_{\log x_{ij}}\right)^2\right). \qquad (6)
$$

Thus, the algorithm is given as follows:

Step 1: From the taxonomy abundance of s samples, we calculated $\mu_{\log x_{ij}}$, $\omega_i$ and $\omega_j$.

Step 2: To infer the correlation $\rho_{ij}^{(k)}$ in the network for kth sample, we substitute $\mu_{\log x_{ij}}$, $\omega_i$ and $\omega_j$ into Eq. (6).

Step 3: We iteratively calculate the correlation coefficients for each sample.

**Prediction of responders in change of HFC.** In the supervised prediction process, we used baseline samples from each group to predict the responders from low/non-responders after intervention. The analysis takes three steps.

Step 1: Through the Single SparCC algorithm, we generated one SSN for each individual in the n samples. We then converted these networks into an edge matrix, which contained the variables consisting of 3321 edges. The edge weight in n samples was taken as the feature.

Step 2: We used the N-sample edge matrix as the input set. The level of HFC after the intervention was the dependent variable, which was used to train an Elastic-Net model.

Step 3: Based on the predicted and true values from step 2, we calculated the prediction metrics. Thereafter, responders from low/non-responders according to their change of HFC after intervention were identified.

In addition, we performed a regression analysis to use the Single SparCC network edges to assess whether they predict HFC change after intervention by using the stat_poly_eq function in the R package 'ggpmisc'.

In an unsupervised prediction process, we directly used the network attributes (edge number, mean degree) of a Single SparCC network (before intervention) to predict the responders from the low/non-responders. A ROC curve was then drawn to visualize the predictive effect (roc function in R package pROC).

**Statistical analysis.** Before statistical test for relative abundance data, taxonomy that appeared in less than 20% of the samples were excluded. Alpha diversity (Shannon index) was performed using the phyloseq (v1.33) package in R v4.0. For the comparison of Shannon index and the absolute content of microbiome, general linear model of analysis of variance controlled for covariates (ANCOVA) for repeated measures (2 factor interactions: group x time) and controlled for change of body weight, baseline value and intervention duration followed by Sidak correction for multiple comparison between the groups. Contrast results (K Matrix) were used to localize the group differences: $*p < 0.05$, $**p < 0.01$ and $***p < 0.001$ by contract to the NI group. Beta diversity analyses were performed to assess the community membership (weighted UniFrac distance for bacterial relative abundance) and functional composition (bray curtis distances for KEGG pathway). Pairwise comparisons of beta diversity between groups were performed using pairwise.perm.manova (R package RVAideMemoire v0.9). $p < 0.05$ is considered as significant in all the above analyses. The Spearman correlation analysis between microbiome and clinical parameters was performed with MATLAB R2020a in samples with complete data. Co-occurrence network analysis was conducted for samples before and after intervention in all 4 groups using pairwise Spearman correlations in MATLAB R2020a, and only the significant correlations ($p < 0.05$) were used for network construction. The networks in each intervention group (before and after intervention) were merged together to facilitate direct comparison and were further visualized in Cytoscape software (v3.8.1). To detect differential taxonomy, KEGG pathway analysis and linear discriminant analysis effect size (LEfSe v1.0, which is an algorithm for high-dimensional biomarker discovery and identifies genomic features such as genes, pathways or taxa as well as characterizing the differences between two or more biological conditions or classes) was performed with cut-offs (LDA Score > 2.0, $p < 0.05$).

**Reporting summary.** Further information on research design is available in the Nature Research Reporting Summary linked to this article.

## Data availability
The study protocol has been published previously (https://doi.org/10.1186/1471-2458-14-48). The raw Illumina sequence data including 16 s rRNA sequence and metagenomics generated in this study have been deposited to the NCBI database under BioProject accession no. PRJNA757939 (https://www.ncbi.nlm.nih.gov/bioproject/?term=PRJNA757939). The data is available for academic use under controlled access in compliance with the regulation of the Ministry of Science and Technology (MOST) of China for the deposit and use of human genomic data. All individual de-identified participant data that support the findings of this study are available from the corresponding author upon reasonable request based on the rule of data protection and consent. The access to the controlled data will be valid for one year from the time of the data accessibility approved. The processed data are available within the Source Data file.

## Code availability
Codes used for computing population and individual networks are available on GitHub (https://github.com/crtsjtu/AELC).

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

## Acknowledgements

Funding to S.C. was provided by the China State Sport General Administration (2013B040, 2015B039), the National Natural Science Foundation of China (Nos. 31571219), the Shanghai Jiao Tong University Zhiyuan Foundation (CP2014013), and to S.L. the Interdisciplinary Program of Shanghai Jiao Tong University (ZH2018QNB05). Funding to L.C. was supported by the National Key R&D Program of China (No. 2017YFA0505500), Strategic Priority Research Program of the Chinese Academy of Sciences (No. XDB38040400), National Natural Science Foundation of China (Nos. 31930022, 31771476, 12026608). Funding to C.Z. was supported by the National Natural Science Foundation of China (Nos. 31930022 and 81871091).In addition to the authors, we thank Mengya Lin for contribution to the absolute microbial abundance measurement. We would like to thank the study team members and participants for their contributions to the success of this trial.

## Author contributions

S.C. designed the study and has full access to all of the data in the study and takes full responsibility for the integrity of the data and for the accuracy of the data analysis. S.C., C.Z., S.L., J.G., D.L. participated in subject recruitment, measurements and intervention process. Y.Y., R.C., S.L. and X.Y. processed fecal samples and the 16 s rRNA sequence analysis. L.W. and L.C. involved in the metagenomics data analysis. X.Z. performed SCFA analysis. R.C., L.W., Y.Y. and T.X. performed the statistical analyses. L.X., P.W., C.Z., L.C. and S.C. checked data analyses and interpretation of the results. R.C., L.W. and S.C. drafted the manuscript. L.X., P.W., C.Z., L.C. and S.C. performed critical revision of the manuscript. All authors have approved the final submission version of the manuscript.

## Competing interests

The authors declared no conflict of interest.

## Additional information

[1]Exercise Translational Medicine Center, Shanghai Center for Systems Biomedicine, Shanghai Jiao Tong University, Shanghai, China. [2]Exercise, Health and Technology Center, Faculty of Physical Education, Shanghai Jiao Tong University, Shanghai, China. [3]School of Life Sciences and Biotechnology, Shanghai Jiao Tong University, Shanghai, China. [4]Key Laboratory of Systems Biology, Shanghai Institute of Biochemistry and Cell Biology, Center for Excellence in Molecular Cell Science, Chinese Academy of Sciences, Shanghai, China. [5]Faculty of Sport Sciences, University of Jyväskylä, Jyväskylä, Finland. [6]School of Physical Education and Training, Shanghai University of Sport, Shanghai, China. [7]Ningbo University, School of Medicine, Ningbo, China. [8]Shidong Hospital of Yangpu District, Shanghai, China. [9]School of Kinesiology, Shanghai University of Sport, Shanghai, China. [10]State Key Laboratory of Microbial Metabolism, School of Life Sciences and Biotechnology, Shanghai Jiao Tong University, Shanghai, China. [11]Key Laboratory of Systems Health Science of Zhejiang Province, Hangzhou Institute for Advanced Study, University of Chinese Academy of Sciences, Hangzhou, China. [12]Guangdong Institute of Intelligence Science and Technology, Zhuhai, China. [13]These authors contributed equally: Runtan Cheng, Lu Wang. ✉email: zhangchenhong@sjtu.edu.cn; lnchen@sibs.ac.cn; sulin.cheng@jyu.fi

