## [Peer Review File · Nature Communications]

REVIEWER COMMENTS

Reviewer #1 (Remarks to the Author):

Thank you for giving me the opportunity to review this manuscript NCOMMS-21-06895 titled: "Response of microbiome network to exercise and diet intervention affect improvement of nonalcoholic fatty liver disease".

In this original manuscript, the authors describe the effect of two types of intervention: 1) aerobic exercise and diet and 2) exercise and diet alone on gut microbiota in patients with non-alcoholic fatty liver disease (NAFLD) and pre-diabetes.

Minor revisions:

The manuscript is well written, and the topic is timely. However, there are some suggestions for the authors to consider:

In the first part of the results section:

- 1) Please define the acronym 'HFC'
- 2) Add the corresponding 'p value' to each result
- 3) Please emphasize that the data on HFC have been previously published
- 4) The authors should explain how they choose the 5% decrease in HFC as a cutoff for stratifying the responders and non-responders.
- 5) The authors should discuss the effect of the intervention on body weight and the association between weight loss and change in gut microbiota. In addition, the authors should investigate whether body weight or weight loss were responsible for the different response to the intervention (responders vs non responders)

In the results section paragraph: "Changes of structure and function of gut microbiota induced by exercise and/or dietary intervention of participants with comorbidity of NAFLD and prediabetes", the authors state that 'These results indicate that microbial diversity deteriorates with the progression of NAFLD, while exercise and dieting may help to maintain the diversity of the gut microbiota'. Please explain how the progression of NALFD was followed-up. Did the authors measure fibrosis? If the authors did not measure fibrosis, they should refer to increase or decrease in HFC and not to NAFLD progression. In the same section, please define the acronym 'LefSe'.

In the methods, please add the description of the intervention in the diet, exercise and diet and exercise groups.

In the primary outcome paper titled: "Effect of aerobic exercise and diet on liver fat in pre-diabetic patients with non-alcoholic-fatty-liverdisease: A randomized controlled trial" the authors explain that the duration of the intervention 'ranged from 6.8 to 11 months.' How many patients completed the study at 6 months and how many at 11 months? Were there any differences in the duration of the intervention between the responders and non-responders? How the different duration of the intervention affected the change in gut microbiota?

Reviewer #3 (Remarks to the Author):

I have some comments mainly on the statistics aspects.

1. It would be useful to include a table to summarize demographics and clinical variables. This table can

be included in the supplementary material.

2. In the results section, please report in the first paragraph, the number of participants in the first groups, how missing data were dealt with, and a consort flow chart of the patients meeting eligibility and included in the study.

3. Page 4, "Changes of structure and function.." The authors stated that "the diversity of gut microbiota was significantly decreased...", please report the associated test statistic value and the p-value. Similar comments to the other line in the same paragraph, "the significant difference between NI and the other groups", please report not only the p-value but also the statistical test, and the observed value for the test statistic.

3. Spell out LefSe analysis and explain what this is this meant.

4. Partial Spearman correlation analysis: please report the number of samples alongside each r and p.

5. Page 6 "Personalized gut microbial networks...". Please report the sensitivity and specificity alongside the AUC value.

6. Supplementary material - Single SparCC network analysis.

Define all math notation used in the formulas.

7. Supplementary material - statistical analysis.

State clearly what are you testing for in the t-test, what is the endpoints that used in the t-test?

Reviewer #4 (Remarks to the Author):

The authors present a good body of research on the impacts of exercise and dietary intervention on NAFLD and prediabetes. It is a novel piece of work that demonstrates the usefulness of non-medical interventions in the treatment of chronic disease. I do feel this point should be discussed more in the manuscript.

Some concerns:

1) did the patients answer a quality of life questionnaire: would be interesting to see if they felt better post intervention

2) was the exercise monitored? did the patients increase their fitness levels? What was compliance level?

3) similarly for diet only lunch was provided; what was compliance to the diet? was other food intake recorded and included in the analysis; this would be very interesting?

4) 8.6 months is a very random intervention time.. how was this chosen?

5) Abstract is poor and needs to be rewritten?

6) Closing paragraph of discussion needs to be better; for example how diet and exercise can be used as a non chemical non medical treatment that improves not only disease status but also gut and overall health. Also financially better.

minor

1) ensure a grammatical check throughout

2) use 16S rRNA not 16s

3) expand HFC in text

Reviewer #5 (Remarks to the Author):

The manuscript by Cheng et al. investigates how gut microbiotas in NAFLD patients respond to exercise and diet in a previously-published interventional trial that showed effective but variable improvement of hepatic fat content (HFC) after the interventions. The authors obtained 16S rRNA compositions from the 76-subject cohort, and profiled metagenomics from a subset of the cohort. They subjected these microbiome data to analyses of 1) compositional changes focused on overall diversity and specific ASVs; 2) correlations of ASV abundance changes with clinical biomarkers; 3) variations of abundances of functional pathways in different groups; 4) co-occurrence network of ASVs and the derived CAGs before and after interventions, and 5) single SparCC network construction based on ASVs and metagenomics data, and the predictive power of these individual baseline networks for the HFC response to interventions.

In the opinion of the reviewer, the paper presents limited advance in our understanding of how gut microbiota contributes to the NAFLD morbidity condition. The work includes different kinds of bioinformatic/statistical analyses, trying to identify associations between microbiome features (ASV, pathway, CAG, single network) and clinical biomarkers (mostly HFC). However, given the small sample size, microbiota variability between subjects and the unaddressed clinical parameters (e.g., age, BMI, NAFLD severity, etc) in each individual, it is uncertain how many of the identified microbiome associations are simply spurious. As stated in their Discussion, some of their findings are contradictory with previous studies and thus inconclusive. Without further experimental evidence (which is technically possible given the existing NAFLD animal models, see Giljae Lee et al. 2020 Nat. Commun.), the findings presented in the work are likely of limited value to the broader field.

Another criticism from the reviewer is the justification of the various analyses (co-occurrence network, single SparCC network) performed in the study. For example, the authors attempted to use a seemingly complicated single SparCC network to predict the intervention efficiency, after the undesired performance of using differentially altered ASVs abundance (this claim also lacks a supporting figure, by the way). However, the logic is largely missing behind why the authors chose to do so instead of exploring and comparing with what people are already doing (e.g., using microbiome compositional changes represented by PCoA eigenvalues). not to mention that the method in its current description lacks sufficient details for other researchers to evaluate and implement (in the Methods section how equation 3, variables a and b are derived is not described; relevant packages are not available).

Other main comments:

This study is based on microbiome compositional relative abundance data. However, numerous recent studies have shown that relative abundance data could be misleading. If possible, the authors should try to measure the absolute abundance of the microbiome samples using some valid methods (e.g. qPCR, flow cytometry, Spikeseq, etc).

Page 3, stratification using a threshold of 5% HFC decrease may need justification as this greatly affects downstream analyses. Is this a commonly accepted standard, or 5% HFC decrease is known to be beyond the normal variability within individuals?

The authors identified multiple ASVs associated with interventions and annotated them with bacterial genera. While the taxonomic resolution of 16S ASV is limited, given the availability of metagenomics data, the authors should be able to map these ASVs to species level. In addition, in analyzing the microbial taxa associations, the authors should also perform analyses in the genus level. Otherwise, claims in the Discussion regarding *Bacteroides* abundance associations are unfounded, as individual ASV abundances do not represent the whole genus.

In Fig 3, the authors claimed the analyses of KEGG pathways were between before and after intervention. However, in Page 5 description, it seems that the analyses only compared between groups after intervention. Given the microbiota diversity between individuals, the baseline profile before intervention should be used as control, instead of the NI group.

Page 6, regarding using individual characteristics to predict intervention efficiency, the authors should compare the predictive power of microbiome features with that of clinical parameters (e.g., age, BMI, NAFLD severity, etc). Given the availability of both 16S ASVs and metagenomics data, which and how the authors chose to use in the single SparCC network needs to be justified. If they both show similar results, the authors should present the more statistically-confident result in the main text.

Discussion section in its current format seems lengthy and out of focus. The authors should also discuss about the biological meaning of the identified associations of networks with clinical outcomes. Importantly, the authors stated that “the personalized prediction here requires the data of other individuals as references, rather than only the independent subject.” In the opinion of the reviewer, this argues against the practical applicability of using individual gut microbial characteristics to predict response of intervention.

Data availability in the current format does not meet the “minimum dataset” that is necessary to interpret, verify and extend the research in the paper. The authors did not provide an accession code of the microbiome data, methodology details and bioinformatics codes.

Minor comments:

Page 3, “it may be worthwhile to analyze the microbiota correlation network at the community level”. The word “community” could be misleading as it is often used to refer to microbial community.

Page 4, “Noticeably, ASVs from the same family or genus showed different behaviours.” The phrase “ASVs from the same family or genus” is misleading given the context.

Page 7, “keystone taxa became divergent but remain stable” – this sentence is unclear.

Page 7, Ref 41 seems wrong.

Page 7, “when using differentially altered ASVs abundance to predict the change of HFC, the diet intervention was the strongest predictor while the exercise did not associate with change of HFC.” – this sentence needs reworking.

Page 8, “we speculate that what exercise intervention improves the microbiota is not its diversity, but the stability of the interaction network between the gut microbiota ecosystem, thereby improving the metabolism of the host.” – the logic leading to the claim of the host metabolism is unfounded.

In the Methods section, 16S rRNA Sequence analysis is confusing in how the authors processed the sequencing data. For example, did the authors actually use QIIME v1.8.0 or QIIME 2?

Table S1 should include more details beyond the core ASV. The authors should report their ASV-level OTU table and the associated CAG assignments in supplemental tables.

REVIEWER COMMENTS

Reviewer #1 (Remarks to the Author):

Thank you for giving me the opportunity to review this manuscript NCOMMS-21-06895 titled: “Response of microbiome network to exercise and diet intervention affect improvement of nonalcoholic fatty liver disease”.

In this original manuscript, the authors describe the effect of two types of intervention: 1) aerobic exercise and diet and 2) exercise and diet alone on gut microbiota in patients with non-alcoholic fatty liver disease (NAFLD) and pre-diabetes.

Minor revisions:

The manuscript is well written, and the topic is timely. However, there are some suggestions for the authors to consider:

Our response: We thank the reviewer for the positive comments. Please see our rebuttal on each point below.

In the first part of the results section:

1) Please define the acronym ‘HFC’

Our response: Done.

2) Add the corresponding ‘p value’ to each result

Our response: Done.

3) Please emphasize that the data on HFC have been previously published

Our response: Done.

4) The authors should explain how they choose the 5% decrease in HFC as a cutoff for stratifying the responders and non-responders.

Our response: We thank the reviewer for the constructive comments. The 5% decrease in HFC was based on the median value of change in HFC, which was -4.85% after intervention. Therefore, we decided to take the around number -5% as cut-off for the responders and non-responders.

5) The authors should discuss the effect of the intervention on body weight and the association between weight loss and change in gut microbiota. In addition, the authors should investigate whether body weight or weight loss were responsible for the different response to the intervention (responders vs non responders)

Our response: We thank the reviewer for this important suggestion. We did report the effect of the intervention on body weight in our previous paper (reference 21). The AEx and AED groups lost roughly 1 kg of their body weight after intervention which was significant. However, no interactions of group by time were found. No significant changes of body weight were found in the Diet and NI groups. In addition, we reported

that the significant reduction of HFC in the intervention groups remained after controlling for the change of body weight, duration of the intervention and baseline HFC with effect size of $\eta=0.159$ ($p=0.006$, AEx vs. NI $p=0.02$; Diet vs. NI $p=0.017$ and AED vs. NI $p=0.001$, respectively). Regarding whether weight is responsible for the intervention effect, we used baseline weight to classify responders/non responders. The ROC curves (Fig. S4b, d) showed that the overall AUC is 0.47 (AED: 0.59, AEx: 0.42, Diet: 0.36), which is not predictive. Therefore, the limited change of body weight may have no impact on the outcome of this report. It is not likely that the differences between the responders and non-responders in gut microbiota were from the body weight or weight loss.

However, we have controlled body weight, baseline value and intervention duration in the gut microbiota diversity analysis. The significance remained and consequently, the result related to this has been added in as follows (page 4, line 13-18):

“We found that the alpha diversity (Shannon index) of gut microbiota was significantly decreased in the NI group ($p = 0.045$) with large individual variance by contrast to the intervention groups, which maintained their diversity after the intervention (Fig. 2a, NI vs. AED $p = 0.011$, NI vs. AEx $p = 0.007$ and NI vs. Diet $p = 0.025$, respectively, analysis of variance with repeated measures adjusted for change of body weight, baseline value and intervention duration).”

For your information, there are no differences in body weight and weight change between responders and non-responders (see results below).

Variables	Group	Mean	SD	p
Weight baseline	Responders	68.4	8.8	0.644
	Non-responders	69.4	10.4	
Weight follow-up	Responders	66.9	9.0	0.502
	Non-responders	68.4	10.5	
Weight change (%)	Responders	-2.19	3.39	0.308
	Non-responders	-1.46	2.76	

In the results section paragraph: “Changes of structure and function of gut microbiota induced by exercise and/or dietary intervention of participants with comorbidity of NAFLD and prediabetes”, the authors state that ‘These results indicate that microbial diversity deteriorates with the progression of NAFLD, while exercise and dieting may help to maintain the diversity of the gut microbiota’. Please explain how the progression of NALFD was followed-up. Did the authors measure fibrosis? If the authors did not measure fibrosis, they should refer to increase or decrease in HFC and not to NAFDL progression.

Our response: We apologize for using the wrong word “progression”. We have now changed the “progression” to “increased HFC”.

In the same section, please define the acronym ‘LEfSe’.

Our response: Done, “linear discriminant analysis effect size for LEfSe” has been added in.

In the methods, please add the description of the intervention in the diet, exercise and diet and exercise groups.

Our response: We have reported the intervention in details in our previous publications. We also briefly described the groups in the results under **Participant characteristics and gut microbiota research design** section. In light of the comment, we have now deleted the group description from results but added more information of the intervention groups as a new section under the methods as follows:

“**Interventions** (page 10, line 13-26):

The exercise (AEx) group participated in a supervised progressive aerobic exercise training program (such as Nordic brisk walking + stretching and other group exercises). Exercise was performed 2-3 times a week, 30 to 60 min per session with 60% to 75% of the maximum oxygen uptake (estimated from fitness test). Each exercise session included a 5 min warm-up and 5 min cool-down period. The diet (Diet) group received a daily prepared meal (lunch), which accounted for 30-40% of the total daily energy intake on the basis of each individual’s dietary intakes and body weight. The meal included 37-40% carbohydrate with 9-13g as fibre, 35-37% fat (SAFA 10%, MUFA 15-20%, PUFA 19 10%) and 25-27% protein plus 5g of soluble fibre (dietary water-soluble fibre). The lunch box was then delivered to the study district community office where the study subjects were gathered. If the participants did not have time to come to pick up the meal, it was delivered to their home and the participants ate it for dinner. The other two meals (breakfast and dinner) were cooked by subjects themselves following the nutritionist advice. In addition, the Diet group was advised to maintain their current level of physical activity during the intervention. The Exercise plus Diet (AED) group performed the same exercise program and followed the same diet as described above for the AEx and Diet groups. The no intervention (NI) group was advised to maintain their current level of physical activity and eating habits during the intervention.”

In the primary outcome paper titled: “Effect of aerobic exercise and diet on liver fat in pre-diabetic patients with non-alcoholic-fatty-liverdisease: A randomized controlled trial” the authors explain that the duration of the intervention ‘ranged from 6.8 to 11 months.’ How many patients completed the study at 6 months and how many at 11 months? Were there any differences in the duration of the intervention between the responders and non-responders?

Our response: We thank the reviewer for the valuable questions. To make it clear, we have now added this information into the methods under **Interventions** section as follows (page 10, line 10-12):

“The duration of the intervention ranged from 6.5 months to 11.1 months and on average 8.6 months. There were no significant differences in the intervention duration among the groups (AEx 8.75 months, Diet 8.6 months, AED 8.5 months and NI 8.6 months) as well as between responders (8.6 months) and non-responders (8,7months).”

How the different duration of the intervention affected the change in gut microbiota?

Our response: Although there were no differences in the duration of intervention among the groups (see methods under **Intervention** section: AEx 8.75 months, Diet 8.6 months, AED 8.5 months and NI 8.6 months), we checked the relationship between duration of intervention and gut microbiota changes. The regression analysis showed that there was actually no relationship between duration of intervention and change of microbiota alpha diversity. For your information, please see the figure below (The R^2 and p values of regression analysis: AED: $R^2 < 0.01$, $p = 0.84$; AEx: $R^2 = 0.046$, $p = 0.36$; Diet: $R^2 < 0.01$, $p = 0.96$; NI: $R^2 = 0.011$, $p = 0.71$). However, there were 7 ASVs abundance (ASV82, ASV356, ASV417, ASV478, ASV1415, ASV1533, ASV2500) which significantly associated with duration of intervention. Since microbiota abundance is a natural sparse matrix and does not conform to the normal distribution, we selected a non-parameter method LEfSe (linear discriminant analysis effect size, an algorithm for high-dimensional biomarker discovery and identifies genomic features such as genes, pathways or taxa as well as characterizing the differences between two or more biological conditions or classes) to compare the ASVs between groups. By this method, we cannot control the impact of duration. Otherwise, in the alpha diversity analysis, we have controlled the duration of intervention, the significant results were remained (see results section page 4 line 13-18).

Reviewer #3 (Remarks to the Author):

I have some comments mainly on the statistics aspects.

1. It would be useful to include a table to summarize demographics and clinical variables. This table can be included in the supplementary material.

Our response: We thank the reviewer for the suggestions. We have now included a supplementary table (Table S1) of demographics and clinical variables of those subjects who have had gut microbiota data measured. The results are presented as marginal mean because group comparison was adjusted for baseline value and intervention duration (See Table S1).

2. In the results section, please report in the first paragraph, the number of participants in the first groups, how missing data were dealt with, and a consort flow chart of the patients meeting eligibility and included in the study.

Our response: Originally, the number of participants was given in the Methods under subjects and design. We have now moved it to the first paragraph of the Results section. In addition, we have changed the order of the text to make it more logical in **Participant characteristics and gut microbiota research design**. Now this section reads as follows (page 3, line 23 to page 4, line 9):

“This study was an 8.6-month, four-arm, randomized trial (Fig. 1a) and participant characteristics have been reported in our previous publication.²¹ Briefly, participants were recruited from 7 health clinical service centers in the Shanghai Yangpu district. They were randomized into four groups: aerobic exercise intervention (AEX, n=29), fiber-enriched low-carbohydrate diet intervention (Diet, n=29), aerobic exercise combined with diet intervention (AED, n=29) and no intervention without guided exercise and dietary intake (NI, n=28).

Since not all the subjects responded to the intervention in a similar manner, we stratified the participants into responders (hepatic fat content (HFC) decreased more than 5%) and low/non-responders (HFC decrease less than 5% or increased) (Fig. 1b). Of all participants, 85 individuals completed the intervention trial. Seventy-six subjects provided paired stool samples at baseline and after the intervention for 16s rRNA gene sequencing. The demographics and clinical variables of those who have had gut microbiota results are presented to Table S1. In addition, to better understand the intervention impact on the function of microbiome, we selected a subset of the cohort (n=42) with the best or worst response, in terms of HFC reduction, to the intervention, and analyzed gut microbiota by the metagenomics data (Fig. 1c).

We previously found that HFC was significantly reduced in the exercise AEx (-24.4%), diet (-23.2%), and AED (-47.9%) groups by contrast to the 20.9% increase in the NI group ($p < 0.001$ for all) after intervention²¹. Of note, 91% of the subjects in the AED group decreased their HFC, and the corresponding figures were 68% in the AEx group, and 86% in the Diet group. In contrast, 72% in the NI group increased their HFC during the intervention period. However, no significant remission or progression of prediabetes was found between the intervention and NI groups based on the glycosylated hemoglobin A1c (HbA1c).²¹ These results indicated that our intervention was effective mainly for HFC reduction.

The microbiota data and clinical parameters were used to (1) characterize the change of gut microbiota composition in response to interventions; (2) determine associations of the microbiome with the level of physical fitness, fat mass, serum biomarkers and short-chain fatty acids (SCFAs); (3) analyze the metabolic function shift of the microbiome in response to interventions; (4) discover the co-occurrence network features of the microbiome ecosystem before and after intervention; and (5) predict personalized response to intervention based on the baseline gut microbiota composition and network.”

3. Page 4, "Changes of structure and function." The authors stated that "the diversity of gut microbiota was significantly decreased...", please report the associated test statistic value and the p-value. Similar comments to the other line in the same paragraph, "the significant difference between NI and the other groups", please report not only the p-value but also the statistical test, and the observed value for the test statistic.

Our response: The p-values and related statistical test have now been added in the related results (page 4, line 13-18) and **Statistical analysis** in Methods section (page 13, line 25-34):

In page 4, line 13-18:

“We found that the alpha diversity (Shannon index) of gut microbiota was significantly decreased in the NI group ($p = 0.045$) with large individual variance by contrast to the intervention groups, which maintained their diversity after the intervention (Fig. 2a, NI vs. AED $p = 0.011$, NI vs. AEx $p = 0.007$ and NI vs. Diet $p = 0.025$, respectively, analysis of variance with repeated measures adjusted for change of body weight, baseline value and intervention duration).”

In page 13, line 25-34

“Before statistical test for relative abundance data, taxonomy that appeared in less than 20% of the samples were excluded. Alpha diversity (Shannon index) was performed using the phyloseq (v1.33) package in R v4.0. For the comparison of Shannon index and the absolute content of microbiome, general linear model of analysis of variance controlled for covariates (ANCOVA) for repeated measures (2 factor interactions: group x time) and controlled for change of body weight, baseline value and intervention duration followed by Sidak correction for multiple comparison between the groups. Contrast results (K Matrix) were used to localize the group differences: * $p < 0.05$, ** $p < 0.01$ and *** $p < 0.001$ by contract to the NI group. Beta diversity analyses were performed to assess the community membership (weighted UniFrac distance for bacterial relative abundance) and functional composition (bray curtis distances for KEGG pathway). Pairwise comparisons of beta diversity between groups were performed using pairwise.perm.manova (R package RVAideMemoire v0.9).”

3. Spell out LefSe analysis and explain what this is this meant.

Our response: LefSe analysis was spelled out in the statical analysis section. Now we have added also in the result (page 4, line 29) and a short explanation as below in the statistical analysis section (page 13, line 40-43):

“... linear discriminant analysis effect size (LEfSe v1.0, which is an algorithm for high-dimensional biomarker discovery and identifies genomic features such as genes, pathways or taxa as well as characterizing the differences between two or more biological conditions or classes)

4. *Partial Spearman correlation analysis: please report the number of samples alongside each r and p.*

Our response: The number of samples was the same for each r and p which has been added in the related partial spearman correlation results in page 4, line 44.

5. *Page 6 "Personalized gut microbial networks...". Please report the sensitivity and specificity alongside the AUC value.*

Our response: The sensitivity and specificity have been added in the results section “**Personalized gut microbial networks ...**” (page 6, line 26-30) as follows:

“.....with a receiver operating characteristic (ROC) area under the curve (AUC) of 0.65 (specificity = 0.58, sensitivity = 0.67). No significant prediction power for the classification in the AED group (AUC = 0.53, specificity = 0.40, sensitivity = 0.67) and the AEx group (AUC = 0.52, specificity = 0.50, sensitivity = 0.60) was found.”

6. *Supplementary material - Single SparCC network analysis. Define all math notation used in the formulas.*

Our response: We have clearly defined and described all math notation for **Single SparCC network analysis** in Methods section (page 12, line 28-29).

“where i represents the i th CAG. To ensure the results were stable, we repeatedly remove the nodes in each network for 1000 times. The median fragility statistics of the 1000 curves were taken as the coordinates for the final curve to plot the robustness curve.”

.....

“Suppose the sample size is S and the number of nodes is D . Letting $a = 2D - 3$, $b = \frac{1}{2(D-1)(D-2)}$, we achieve an analytical solution for ρ_{ij} ”.

7. *Supplementary material - statistical analysis.*

State clearly what are you testing for in the t-test, what is the endpoints that used in the t-test?

Our response: We apology for the mistake in the previous version. To clarify this issue, we modified the descriptions in the Methods section **Statistical analysis** (page 13, line 26-31)

“For the comparison of Shannon index and the absolute content of microbiome, general linear model of analysis of variance controlled for covariates (ANCOVA) for repeated measures (2 factor interactions: group x time) and controlled for change of body weight, baseline value and intervention duration followed by Sidak correction for multiple

comparison between the groups. Contrast results (K Matrix) were used to localize the group differences: * $p < 0.05$, ** $p < 0.01$ and *** $p < 0.001$ by contrast to the NI group.”

Reviewer #4 (Remarks to the Author):

The authors present a good body of research on the impacts of exercise and dietary intervention on NAFLD and prediabetes. It is a novel piece of work that demonstrates the usefulness of non-medical interventions in the treatment of chronic disease. I do feel this point should be discussed more in the manuscript.

Some concerns:

1) did the patients answer a quality of life questionnaire: would be interesting to see if they felt better post intervention

Our response: We thank the reviewer for the comments. Unfortunately, we do not have the quality-of-life questionnaire onset of the study.

2) was the exercise monitored? did the patients increase their fitness levels? What was compliance level?

Our response: Yes, the exercise was monitored by heart rate and step counts. The exercise was a supervised progressive aerobic exercise training program (see Methods under **Interventions** section (page 10, line 13-26). The fitness level indicated by VO₂max was increased significantly in AEx and AED (see the Fig. 1a). The compliance level of the AEx and AED for the exercise days was on average 12135 steps and energy expenditure (EE) of 384 kcal/day, while during other days it was 8043 steps and EE 250 kcal/day.

The compliance of participation of the study was reported in our early report (reference 21) as: “The overall compliance was 62.1% for the NI group, 75.9% for the AEx group, 78.6% for the Diet group and 79.3% for the AED group, respectively. The main reasons for drop-out were the following: family reasons (n=6); diseases not related to the present trial (n=5); loss of interest (n=5); travel (n=6) and other personal reasons (n=6). The high drop-out rate in the NI group was mainly due to loss of interest (disappointment at not being in the intervention groups).” Therefore, we did not repeat this information in this report.

3) similarly for diet only lunch was provided; what was compliance to the diet? was other food intake recorded and included in the analysis; this would be very interesting?

Our response: Detailed information regarding the diet intervention has been given in reference 21 and also brief explanation has been given in Methods **Interventions** section (page 10, line 16-23). We have explained that

“The diet (Diet) group received a daily prepared meal (lunch), which accounted for 30-40% of the total daily energy intake on the basis of each individual’s dietary intakes and body weight. The meal included 37-40% carbohydrate with 9-13g as fibre, 35-37% fat (SAFA 10%, MUFA 15-20%, PUFA 19 10%) and 25-27% protein plus 5g of soluble

fibre (dietary water-soluble fibre). The lunch box was then delivered to the study district community office where the study subjects were gathered. If the participants did not have time to come to pick up the meal, it was delivered to their home and the participants ate it for dinner. The other two meals (breakfast and dinner) were cooked by subjects themselves following the nutritionist advice. In addition, the Diet group was advised to maintain their current level of physical activity during the intervention.”

The overall compliance was 78.6% for the Diet group as we have explained in our earlier report (reference 21). The food intake records were also reported in reference 21. We did not include the associations of specific micronutrients analysis with certain keystone taxa of gut microbiota as this was not the purpose of this study.

4) 8.6 months is a very random intervention time. how was this chosen?

Our response: Originally, we have planned to have 6 months intervention and 6 months follow-up (see from our provided file of the study protocol). Due to the conflicts of time schedule for hepatic fat content measurement in the clinical MRI centre at the end of intervention (the MRI scanner was loaded for pre-scheduled health checking for more than 2 months); thus, we were not able to perform the scans at the onset of the intervention for part of the participants. In addition, some scheduled times have to be postponed to give priority for the clinical patients. Therefore, we kept the intervention ongoing till all the participants have reached 6 months in the intervention. The other reason was the limit funding which we have explained early. Therefore, the duration of the intervention ranged from 6.5 months to 11.1 months and on average 8.6 months.

5) Abstract is poor and needs to be rewritten?

Our response: The abstract has been modified according your suggestion.

6)Closing paragraph of discussion needs to be better; for example how diet and exercise can be used as a non chemical on medical treatment that improves not only disease status but also gut and overall health. Also financially better.

Our response: We thank the reviewer for this suggestion. We have modified the last paragraph of the discussion as follows (page 9, line 26-28):

“Using the network features of gut microbiota to predict individual response to exercise and diet interventions may help to choose more precise and effective treatment for different NAFLD pateints.”

minor

1) ensure a grammatical check throughout

2) use 16S rRNA not 16s

3) expand HFC in text

Our response: All these points have been corrected.

Reviewer #5 (Remarks to the Author):

The manuscript by Cheng et al. investigates how gut microbiotas in NAFLD patients respond to exercise and diet in a previously-published interventional trial that showed effective but variable improvement of hepatic fat content (HFC) after the interventions. The authors obtained 16S rRNA compositions from the 76-subject cohort, and profiled metagenomics from a subset of the cohort. They subjected these microbiome data to analyses of 1) compositional changes focused on overall diversity and specific ASVs; 2) correlations of ASV abundance changes with clinical biomarkers; 3) variations of abundances of functional pathways in different groups; 4) co-occurrence network of ASVs and the derived CAGs before and after interventions, and 5) single SparCC network construction based on ASVs and metagenomics data, and the predictive power of these individual baseline networks for the HFC response to interventions.

In the opinion of the reviewer, the paper presents limited advance in our understanding of how gut microbiota contributes to the NAFLD morbidity condition. The work includes different kinds of bioinformatic/statistical analyses, trying to identify associations between microbiome features (ASV, pathway, CAG, single network) and clinical biomarkers (mostly HFC). However, given the small sample size, microbiota variability between subjects and the unaddressed clinical parameters (e.g., age, BMI, NAFLD severity, etc) in each individual, it is uncertain how many of the identified microbiome associations are simply spurious. As stated in their Discussion, some of their findings are contradictory with previous studies and thus inconclusive. Without further experimental evidence (which is technically possible given the existing NAFLD animal models, see Giljae Lee et al. 2020 Nat. Commun.), the findings presented in the work are likely of limited value to the broader field.

Our response: We thank the reviewer for the careful evaluation of our study. The main purpose of this study was to find gut microbiota characteristics that reflect the influence of intervention on NAFLD through analysis of different levels and methods, rather than analysing the relationship between different characteristics as stated by the reviewer. We found that it may be a more effective approach to investigate NAFLD intervention from the perspective of gut microbiota ecosystem stability, compared to ASV abundance and pathway differences.

It is a debatable how lifestyle intervention could impact the gut microbial ecosystem and thereby affect NAFLD and glucose metabolism impairment. When viewing most of the reports on human studies of intervention and NAFLD (e.g.

<https://pubmed.ncbi.nlm.nih.gov/30667502/>, n = 40, t = 8 weeks;

<https://pubmed.ncbi.nlm.nih.gov/30142427/>, n = 140, intervention group n = 70, t = 12 months; <https://pubmed.ncbi.nlm.nih.gov/27521509/>, n = 24, t = 12 weeks;

<https://pubmed.ncbi.nlm.nih.gov/21708823/>, n = 19, t = 8 weeks;

<https://pubmed.ncbi.nlm.nih.gov/22213436/>, n = 18, t = 16 weeks.

And intervention on gut microbiota: <https://pubmed.ncbi.nlm.nih.gov/29097438/>, n = 60, t = 8 weeks; <https://pubmed.ncbi.nlm.nih.gov/32075887/>, n = 82, t = 8 weeks; <https://pubmed.ncbi.nlm.nih.gov/30425247/>, n = 60, t = 16 weeks), our sample size is in the range of those reports. Importantly, most of those reports have only two arms and relatively short intervention duration. There are only a very few studies which have lifestyle intervention on gut microbiota in NAFLD with no mention of comorbidity. Our study is relatively long and has 4-arm well controlled interventions. The small sample in the metagenomics was used to partly validate the results obtained by *16S rRNA*, of which the sample size is comparable with most of the studies. Our study with relatively long-term intervention is one major strength, which provides both statistical and sample-specific network insights into the complex gut microbial ecosystem in patients with NAFLD and glucose metabolism impairment.

Regarding the reviewer's concern "*the unaddressed clinical parameters (e.g., age, BMI, NAFLD severity, etc) in each individual, it is uncertain how many of the identified microbiome associations are simply spurious.*" There are no differences in age, body weight, BMI, and HFC (indicate the NAFLD severity) for different groups (see reference 21 and Table S1). In our correlation analysis, we did control the body weight and fat mass. Age would not affect the intervention effects on gut microbiota due to no age different among the groups. Thus, it is likely that those factors would affect the identified microbiome associations.

Regarding the reviewer's concern "*As stated in their Discussion, some of their findings are contradictory with previous studies and thus inconclusive. Without further experimental evidence (which is technically possible given the existing NAFLD animal models, see Giljae Lee et al. 2020 Nat. Commun), the findings presented in the work are likely of limited value to the broader field.*" It is not unusual that the results from different studies are contradictory or inconclusive. In this particular study, we explained that the different observations regarding *Bacteroides* abundance after intervention have already been reported early and these inconclusive results may be due to differences in ethnicity, living environment and lifestyle among the cohorts. However, our core results are novel and in line with current understanding of human gut microbiota and NAFLD. We did performed analysis by both traditional statistics and novel approaches. To ensure that our observed results from 16S rRNA are firm, we also used metagenomics to confirm those results.

Our study provides a new perspective on the research of gut microbiota, that is, in addition to focusing on the differential strains themselves, the topological properties of the microbial ecosystem community network are also of great importance and significance. There is a clinical possibility that the gut microbiota can be used to precisely predict individual response to exercise and diet interventions. Hence, using non-pharmaceutical treatment such as exercise and diet for improving disease status and microbial ecosystem, it is better and has lower risk than drug and surgical treatment in the prevention and treatment of the comorbidity condition related to lifestyle, from both overall healthy and economical perspectives.

The study of Giljae Lee et al. 2020 Nat. Commun is indeed important and we have included it in the manuscript as reference 33. Our result of association between *Ruminococcaceae* and HFC is in agreement with their result. However, that study is cross-sectional and they point out in the paper that future longitudinal studies are needed. Our study with relatively long-term and 4-arm well controlled interventions are one of very few existing reports and therefore may have important value to evoke further studies to advance our understanding of how gut microbial ecosystem respond to the lifestyle changes and associated diseases.

Another criticism from the reviewer is the justification of the various analyses (co-occurrence network, single SparCC network) performed in the study. For example, the authors attempted to use a seemingly complicated single SparCC network to predict the intervention efficiency, after the undesired performance of using differentially altered ASVs abundance (this claim also lacks a supporting figure, by the way). However, the logic is largely missing behind why the authors chose to do so instead of exploring and comparing with what people are already doing (e.g., using microbiome compositional changes represented by PCoA eigenvalues). not to mention that the method in its current description lacks sufficient details for other researchers to evaluate and implement (in the Methods section how equation 3, variables a and b are derived is not described; relevant packages are not available).

Our response: To some extent, we do not agree with this criticism. Indeed, we have performed some of the analyses by using traditional analysis methods to assess the diversity and abundance of gut microbial composition (see results under section of **Changes of structure and function of gut ...**, page 4, line 10 - page 5, line 14). We also discussed those results in the discussion section (page 7, line 26 - page 8, line 2).

The justification of the various analyses (co-occurrence network, single SparCC network) performed in the study has been given in our manuscript. Co-occurrence network has been widely used in the microbiota studies recent years. As we explained in the Introduction (page 3, line 5-12), i.e.

“Currently, it is widely accepted that, in the microbial community, the keystone taxa are drivers of microbiome structure and function, and in particular, their interaction network, which plays an important role in microbial functions and disease progression.²⁰ Therefore, to establish an effective intervention strategy, it may be worthwhile to analyze the microbiota correlation network at the microbial community level, while assessing the physiological mechanisms at the individual level and further exploring individual microbial networks that underlie the differences between responders and low/non-responders of various interventions.”

In the results under section **Intervention induced change in gut microbiota co-occurrence network:**

Page 5, line 30-34:

“It has been shown that bacterial species in the human gut may survive, adapt, and decline as interdependent functional groups (guilds) responding to environmental perturbations.^{24,25} To identify bacteria in the gut ecosystem that responded as functional groups to interventions, we adopted "co-abundance groups (CAGs)" to analyze the community structure in the microbial ecosystem.²⁴” and

Page 6, line 4-11:

“Because the complexity caused by the nonlinear dynamics of the nodes is related to the network architecture, we assessed the distribution of the network degree to judge the topological features of the microbial network (Fig. S2b). We found that most of the connections of the co-occurrence network were concentrated in a few nodes, which is an important features of a scale-free network. This kind of network with a power-law distribution comprise highly connected nodes, which are defined as hubs.²⁹ The hubs in a microbial network have been proposed as keystone taxa, as their removal has been computationally shown to cause a drastic shift in the composition and functioning of a microbiome.²⁰”

In the discussion section (page 8, line 21-25), we specifically discussed the advantage of using co-occurrence network and the novel findings by using this method. We believe that our argument is sound and solid based on our results and current knowledge “...what exercise intervention improves not the gut microbiota diversity, but the stability of the interaction network of the gut microbiota ecosystem, thereby improving the metabolism of the host. The keystone taxa determined by the microbiota population network topology may be a new perspective beyond finding key bacteria by abundance and diversity comparison.”

The justification of using single SparCC network was explained in the results under **Personalized gut microbial networks can predict the intervention efficiency of individual patients** section (page 6, line 30-32) as

“Second, since ASVs abundance could not describe the individual network characteristics, we developed a Single SparCC network method by combining a SparCC network²⁶ with a sample specific network (SSN)³⁰ (Fig. 5a).”

In the discussion section (page 8, line 36 to page 9 line 16) that

“Our Single SparCC method evaluated not only microbiome composition and abundance, but also the interactions between species. Such a network can be used to quantify the system-wide features, e.g., robustness or stability of the system, and further predict intervention effects. Various studies^{28,43} have adopted the concept of connectivity and robustness from network theory to investigate the rule of microbial community in diseases from an ecological perspective. Actually, there is increasing evidence indicating a link between undesirable health conditions with altered microbial assembly process and a more fragile microbiome network.^{40,44} Our results indicated that the Single SparCC network of gut microbiota is a valid method to predict the responders from low/non-responders for exercise intervention and therefore opens a new avenue to assess the effect of exercise intervention on gut microbiota at an individual level. Interestingly, we found

that when differentially altered ASVs abundance is used to classify the responders and low/non-responders, it has the best prediction power in the diet group but invalid in the exercise group. In terms of the AUC values, the Single SparCC network of gut microbiota was better differential features to responders from low/non-responders than age, body weight and BMI in our study. These results indicate that corresponding gut microbial characteristics can be used to predict responsiveness of a subject to specific intervention methods. This is of great clinical significance in improving effectiveness and efficiency of interventional therapy of NAFLD. Notably, the personalized prediction here requires a number of samples or other individuals measured as the reference samples, against which the network of each test sample can be projected. Comparing with the conventional machine learning methods, although the reference samples can be viewed as the training samples, our method can further extract the second-order statistical information (e.g. correlation or association or network) of each individual approximately, thus providing system-wide features of each test sample. In general, our results are important in terms of how to assess the intervention effect on physiology and pathophysiology and to develop personalized lifestyle treatment for such patients through their individual gut microbiota network.”

The criticism on “*the method in its current description lacks sufficient details for other researchers to evaluate and implement (in the Methods section how equation 3, variables a and b are derived is not described; relevant packages are not available)*” is well taken. The explanation of a and b in equation 3 was given after equation 5 in the previous manuscript. For ease of reading, we have moved its position to the part before equation 3. Single SparCC network is the original method of this work (page 12, line 28-29). The details of the algorithm have been given in the Methods **Single SparCC network analysis** section (page 12, line 10 - page 13, line 8), and there is no reference to the relevant package.

Other main comments:

This study is based on microbiome compositional relative abundance data. However, numerous recent studies have shown that relative abundance data could be misleading. If possible, the authors should try to measure the absolute abundance of the microbiome samples using some valid methods (e.g. qPCR, flow cytometry, Spikeseq, etc).

Our response: We have re-quantitatively extracted the DNA of each sample, and performed qPCR experiments on the V3-V4 region. We found that there is no significant difference before and after the intervention between the groups (Fig. S1a). The total bacterial content of each subject in the same group at two time points also have no significant differences (Fig. S1b-e). The specific results and methods are added into manuscript under “**Changes of structure and function of ...**” section (page 4, line 23-28) as

“To avoid only included the relative abundance data in the analysis which may introduce the bias, we then measured the absolute abundance of microbiome by quantitative real-time PCR (qPCR) targeting the 16S rRNA gene. We found that there were no significant differences between baseline and after intervention as well as among the groups (Fig.

S1a). Moreover, the total bacterial content of each subject in the same group at two time points also did not differ significantly (Fig. S1b-e).”

and **qPCR of total fecal bacteria** (page 10, line 41-49) in Methods section as

“A plasmid of the 16S full-length positive Ruminococcus strain (n = 1010 copies/μl) was diluted according to different gradients successively to 109 , 108 , 107 , 106 , 105, 104 , 103 , and 102 copies/μl. qPCR was performed in a 20-μl reaction system containing template (20 ng), primer Uni331F (5'-TCCTACGGGAGGCAGCAGT-3'), primer Uni797R(5'-GGACTACCAGGGTA TCTAATCCTGTT-3') , and supermix (Bio-Rad) on a qTOWER3G touch system, with 2 replicates for standard and sample DNA. The PCR conditions were 95°C for 5 min, followed by 40 cycles of 95°C for 20 s, 60°C for 60 s, and 80°C for 5 s, and Melting curve 60°C to 95°C. A standard curve was determined through a linear fit of the copy number and CT value of the plasmid in different gradients. The copy number of sample DNA was calculated through a standard curve.”

Page 3, stratification using a threshold of 5% HFC decrease may need justification as this greatly affects downstream analyses. Is this a commonly accepted standard, or 5% HFC decrease is known to be beyond the normal variability within individuals?

Our response: 5% decrease in HFC was based on the median value of change in HFC, which was -4.85% after intervention. Therefore, we decided to take the around number - 5% as cut-off for the responders and non-responders.

The authors identified multiple ASVs associated with interventions and annotated them with bacterial genera. While the taxonomic resolution of 16S ASV is limited, given the availability of metagenomics data, the authors should be able to map these ASVs to species level. In addition, in analyzing the microbial taxa associations, the authors should also perform analyses in the genus level. Otherwise, claims in the Discussion regarding Bacteroides abundance associations are unfounded, as individual ASV abundances do not represent the whole genus.

Our response: We are afraid the reviewer misunderstood the ASVs and taxonomy classification of ASVs. First, new methods have been developed that resolve amplicon sequence variants (ASVs) from Illumina-scale amplicon data without imposing the arbitrary dissimilarity thresholds that define molecular OTUs. ASV methods infer the biological sequences in the sample prior to the introduction of amplification and sequencing errors, and distinguish sequence variants differing by as little as one nucleotide. This new ASV methods are explicitly intended to replace OTUs as the atomic unit of analysis. ASV methods have demonstrated sensitivity and specificity as good or better than OTU methods and better discriminate ecological patterns. Then we did ASV-level analysis means species even strain-level analysis. Taxonomy classification comparing our query sequences (ASVs) to a reference database of sequences with known taxonomic composition. Simply finding the closest alignment is not really good enough because other sequences that are equally close or nearly as close may have different taxonomic annotations. So we use taxonomy classifiers to determine the closest

taxonomic affiliation with some degree of confidence or consensus, which may not be a species name if one cannot be predicted with certainty, because as a common sense, amplicon of 16S rRNA V3-V4 region is only included two hypervariable regions and is not long enough to be classified to a species. If we say “Bacteroides ASV1”, it means ASV1 is a bacterial specie (even a strain, sometimes) belong to Bacteroides.

In Fig 3, the authors claimed the analyses of KEGG pathways were between before and after intervention. However, in Page 5 description, it seems that the analyses only compared between groups after intervention. Given the microbiota diversity between individuals, the baseline profile before intervention should be used as control, instead of the NI group.

Our response: We apologize that it was a mistake in the previous legend of Fig.3 on this issue. In this new version, we have changed ‘between before and after intervention’ into ‘between intervention groups and NI group after intervention’ in the new legend of Fig.3 (page 19, line 2-3).

Page 6, regarding using individual characteristics to predict intervention efficiency, the authors should compare the predictive power of microbiome features with that of clinical parameters (e.g., age, BMI, NAFLD severity, etc). Given the availability of both 16S ASVs and metagenomics data, which and how the authors chose to use in the single SparCC network needs to be justified. If they both show similar results, the authors should present the more statistically-confident result in the main text.

Our response: We thank the reviewer for the good comments. We have now made the prediction analysis by using age, body weight (WT) and body mass index (BMI). This information has been added into the results under **Personalized gut microbial networks can predict the intervention efficiency of individual patients** section and discussed in the discussion as bellow:

In Results (page 6, line 40-45):

“For the purpose of comparison, we also tested the ability of age, body weight (WT) and body mass index (BMI) to distinguish the responsiveness of the interventions (Fig. S4). The result showed that these clinical parameters were not able to differentiate the responders from the low/non-responders (age: AUC = 0.437, p = 0.347; WT: AUC = 0.470, p = 0.655; BMI: AUC = 0.519, p = 0.771), except for BMI in the AED group.”

In Discussion (page 9, line 3-5):

“In terms of the AUC values, the Single SparCC network of gut microbiota was better differential features to responders from low/non-responders than age, body weight and BMI in our study.”

Discussion section in its current format seems lengthy and out of focus. The authors should also discuss about the biological meaning of the identified associations of networks with clinical outcomes.

Our response: In light of the comment, we have modified the Discussion section and added the biological meaning of network connectivity as follows:

In page 7, line 26-35:

“Members from *Ruminococcus* have been reported to produce SCFAs from complex carbohydrates.^{31,32} A recent study³³ showed that *Ruminococcaceae* was negatively correlated with the fibrosis severity. In agreement with this, *Ruminococcaceae* was found negatively correlation with HFC in our study. In addition, some members of *Bacteroides* contribute to the release of energy from dietary fiber and starch, and they are likely to be a major source of propionate.³⁴ Accordingly, we found that *Bacteroides* was positively correlated with propionic acid, isobutyric acid and isovaleric acid. A previous study showed that compared to western countries, the abundance of *Bacteroides* in NAFLD is much lower in Chinese individuals.³⁵ However, our result suggested an important energy-extracting role of *Bacteroides*, although their abundance is relatively low in Chinese population.”

In page 8, line 9-12:

“In combination with these findings, our results indicate that microbiota as an ecosystem were more healthy and stable after intervention. Interestingly, when ASVs abundance is used to predict the change of HFC, it has the best predictive power in the diet group, but is not predictive in the exercise group.”

In page 8, line 26-36:

“No disease-specific drug/medical therapy for NAFLD is available due to its complexly pathogenetic mechanism, and management of lifestyle (such as exercise and diet) in patients is the only efficient intervention in clinical practices. However, studies have shown that not all participants respond to interventions of exercise or diet in a similar way,¹⁶ which may be attributable to the individual-variation and response of gut microbiota in patients. In some studies, 50% of the study participants show no change in their gut microbiota composition after an exercise intervention.¹⁷ Then prediction of improvement of clinical phenotypes in different patients based on their gut microbiota would be important to increase efficiency of NAFLD treatment. In the previous studies, they ascribed to use the group-level changes of bacteria abundance to construct the prediction models, which disregards the fact that the members of gut microbiota show ecological interactions such as competition or cooperation with each other.”

In page 8, line 39-43:

“Various studies^{28,43} have adopted the concept of connectivity and robustness from network theory to investigate the rule of microbial community in diseases from an ecological perspective. Actually, there is increasing evidence indicating a link between undesirable health conditions with altered microbial assembly process and a more fragile microbiome network.^{40,44}”

In page 9, line 46 - page 10, line 13:

“Interestingly, we found that when differentially altered ASVs abundance is used to classify the responders and low/non-responders, it has the best prediction power in the diet group but invalid in the exercise group. In terms of the AUC values, the Single SparCC network of gut microbiota was better differential features to responders from low/non-responders than age, body weight and BMI in our study. These results indicate that corresponding gut microbial characteristics can be used to predict responsiveness of a subject to specific intervention methods. This is of great clinical significance in improving effectiveness and efficiency of interventional therapy of NAFLD. Notably, the personalized prediction here requires a number of samples or other individuals measured as the reference samples, against which the network of each test sample can be projected. Comparing with the conventional machine learning methods, although the reference samples can be viewed as the training samples, our method can further extract the second-order statistical information (e.g. correlation or association or network) of each individual approximately, thus providing system-wide features of each test sample.”

Importantly, the authors stated that “the personalized prediction here requires the data of other individuals as references, rather than only the independent subject.” In the opinion of the reviewer, this argues against the practical applicability of using individual gut microbial characteristics to predict response of intervention.

Our response: We have modified the discussion according to your suggestion (see page 9, line 8-13).

“Notably, the personalized prediction here requires a number of samples or other individuals measured as the reference samples, against which the network of each test sample can be projected. Comparing with the conventional machine learning methods, although the reference samples can be viewed as the training samples, our method can further extract the second-order statistical information (e.g. correlation or association or network) of each individual approximately, thus providing system-wide features of each test sample.”

Data availability in the current format does not meet the “minimum dataset” that is necessary to interpret, verify and extend the research in the paper. The authors did not provide an accession code of the microbiome data, methodology details and bioinformatics codes.

Our response: We have added a **DATA AVAILABILITY** section in the manuscript (page 14, line 2-6) as follows:

“The raw Illumina sequence data including 16s rRNA sequence and metagenomics have been deposited to the NCBI database under BioProject accession no. PRJNA757939. All other data that support the findings of this study are available from the corresponding author upon reasonable request. Source data are provided with this paper.”

Minor comments:

Page 3, “it may be worthwhile to analyze the microbiota correlation network at the

community level”. The word “community” could be misleading as it is often used to refer to microbial community.

Our response: We have corrected to be “microbial community” (page 3, line 9).

Page 4, “Noticeably, ASVs from the same family or genus showed different behaviours.” The phrase “ASVs from the same family or genus” is misleading given the context.

Our response: There is no misleading here. As we explained above, ASVs means bacteria species (even strain-level). “ASVs from the same family or genus” means the bacteria species or strains belong to the same family or genus. As a common sense, that family and genus is taxonomy unit and one family or genus contents lots of different species and strains. Function of bacteria is species/strain specific, which means the bacteria belong to the same family or genus have different function and show different relationship with the host health.

Page 7, “keystone taxa became divergent but remain stable” – this sentence is unclear.

Our response: We have changed this sentence to “keystone taxa became divergent but their connections remained stable.” (page 7, line 21-22)

Page 7, Ref 41 seems wrong.

Our response: This is indeed a wrong reference and we have deleted it.

Page 7, “when using differentially altered ASVs abundance to predict the change of HFC, the diet intervention was the strongest predictor while the exercise did not associate with change of HFC.” – this sentence needs reworking.

Our response: Thank you for your suggestion. We have modified the sentence as follows (page 8, line 11-12):

“when ASVs abundance is used to predict the change of HFC, it has the best predictive power in the diet group, but is not predictive in the exercise group.”

Page 8, “we speculate that what exercise intervention improves the microbiota is not its diversity, but the stability of the interaction network between the gut microbiota ecosystem, thereby improving the metabolism of the host.” – the logic leading to the claim of the host metabolism is unfounded.

Our response: We agree that we cannot claim the direct link for the effect of exercise on the gut microbiota ecosystem, which quantifies how exercise through the microbiota to affect the host metabolism. The texts have been modified as follows (Page 8, line 21-22):

“we speculate that what exercise intervention improves not the gut microbiota diversity, but the stability of the interaction network of the gut microbiota ecosystem, thereby improving the metabolism of the host.”

In the Methods section, 16S rRNA Sequence analysis is confusing in how the authors processed the sequencing data. For example, did the authors actually use QIIME v1.8.0 or QIIME 2?

Our response: We have used QIIME 2 in the analysis and this information has been clarified in the methods (page 11, line 17).

Table S1 should include more details beyond the core ASV. The authors should report their ASV-level OTU table and the associated CAG assignments in supplemental tables.

Our response: As we explained, ASV and OTU are totally different. ASVs resolve exactly, down to the level of single-nucleotide differences over the sequenced gene region and we have thousands ASVs in our dataset. Because ASV is not defined at certain thresholds as OTUs, anyone re-analyze with our raw sequencing data that we have submitted to NCBI will obtain exactly same ASV table as we do. Then, there is no earthly reason to report our ASV-level features table.

REVIEWER COMMENTS

Reviewer #4 (Remarks to the Author):

I am happy that the authors have addressed all my comments.

Reviewer #5 (Remarks to the Author):

The authors' revision has addressed most of the reviewer's concerns. However, several points remain as below.

As stated in the rebuttal, "The main purpose of this study was to find gut microbiota characteristics that reflect the influence of intervention on NAFLD through analysis of different levels and methods, rather than analysing the relationship between different characteristics as stated by the reviewer", the work mostly looks for correlations between microbiota characteristics and interventional outcome and shows predictive powers of personalized baseline microbiome network, rather than demonstrating any causal or mechanistic relationship between microbiota and clinical condition. Therefore, the current title "Response of microbiome network to exercise and diet intervention affect improvement of nonalcoholic fatty liver disease", which states a causal relationship, may not be the most appropriate. The authors may consider low-tuning the conclusion to something like "Response of microbiome network to exercise and diet intervention in nonalcoholic fatty liver disease" or "Personalized microbiome network predicts clinical response to exercise and diet intervention in nonalcoholic fatty liver disease".

The authors may have misunderstood the reviewer's criticism on interpreting the taxonomic level of individual ASVs. Of course, 16S ASV usually gives better specificity than traditional OTU and the limited length of 16S ASV is known to be insufficient for species-level classification. However, given that the study has also performed, for considerable samples that overlap with 16S-sequenced ones, shotgun metagenomics which rendered species-level classification, the authors should be able to combine these two datasets (based on compositional frequencies or patterns of temporal changes for example) and point out which species the highlighted ASVs are likely to be. This would be useful information for others, as simply listing the ASV numbers without detailed information (e.g., exact sequences) would make the conclusions generated in this study difficult for others to evaluate. The authors refused to provide an ASV table as supplemental, stating that "anyone re-analyze with our raw sequencing data that we have submitted to NCBI will obtain exactly same ASV table as we do". In reality, as we know, any difference in sequence processing, software version or system environment could change how the ASVs are numbered, thus making the ASV statements in the study hard to interpret. To obtain species-level call for the ASVs of interest would be important for any future mechanistic studies in this field. Also a minor point on Fig 2c: the annotations on the left column only appear for a few ASVs; do the other ASVs lack confident annotations (maybe not, as the main text discussed about ASV2513), or each annotation covers multiple ASVs?

The authors should also be cautious on their genus/family-level statements (Line 284-297), as all the analyses, as written, were done on the ASV (using 16S data) or species (using metagenomic data) level.

For example, the increase of ASV2077 and ASV2513 alone is NOT equal to the increase of the whole Bacteroides genus, unless only two ASVs were found to be classified as Bacteroides (which seems unlikely). To support such statements on Bacteroides and Ruminococcaceae, the authors would need to perform LEfSe analyses on genus/family level (which is very feasible). If the authors instead decide to stick with ASV/species-level analyses, the conclusion should be modified to accurately reflect the data. In this case, again, having a biologically-meaningful taxonomy like Bacteroides uniformis would make the study much more referenceable than the current ASV1 etc.

REVIEWER COMMENTS

Reviewer #5 (Remarks to the Author):

The authors' revision has addressed most of the reviewer's concerns. However, several points remain as below.

As stated in the rebuttal, "The main purpose of this study was to find gut microbiota characteristics that reflect the influence of intervention on NAFLD through analysis of different levels and methods, rather than analysing the relationship between different characteristics as stated by the reviewer", the work mostly looks for correlations between microbiota characteristics and interventional outcome and shows predictive powers of personalized baseline microbiome network, rather than demonstrating any causal or mechanistic relationship between microbiota and clinical condition. Therefore, the current title "Response of microbiome network to exercise and diet intervention affect improvement of nonalcoholic fatty liver disease", which states a causal relationship, may not be the most appropriate. The authors may consider low-tuning the conclusion to something like "Response of microbiome network to exercise and diet intervention in nonalcoholic fatty liver disease" or "Personalized microbiome network predicts clinical response to exercise and diet intervention in nonalcoholic fatty liver disease".

Our response: We thank the reviewer for the careful evaluation and thoughtful comments. We have modified our title according to the reviewer's suggestion and the new title is:

"Response of microbiome network to exercise and diet intervention in nonalcoholic fatty liver disease"

The authors may have misunderstood the reviewer's criticism on interpreting the taxonomic level of individual ASVs. Of course, 16S ASV usually gives better specificity than traditional OTU and the limited length of 16S ASV is known to be insufficient for species-level classification. However, given that the study has also performed, for considerable samples that overlap with 16S-sequenced ones, shotgun metagenomics which rendered species-level classification, the authors should be able to combine these two datasets (based on compositional frequencies or patterns of temporal changes for example) and point out which species the highlighted ASVs are likely to be. This would be useful information for others, as simply listing the ASV numbers without detailed information (e.g., exact sequences) would make the conclusions generated in this study difficult for others to evaluate. The authors refused to provide an ASV table as supplemental, stating that "anyone re-analyze with our raw sequencing data that we have submitted to NCBI will obtain exactly same ASV table as we do". In reality, as we know, any difference in sequence processing, software version or system environment could change how the ASVs are numbered, thus making the ASV statements in the study hard to interpret. To obtain species-level call for the ASVs of interest would be important for any future mechanistic studies in this field. Also a minor point on Fig 2c: the annotations on the left column only appear for a few ASVs; do the other ASVs lack confident annotations (maybe not, as the main text discussed about ASV2513), or each annotation covers multiple ASVs?

Our response: Thanks for the reviewer's comments. Regarding the reviewer's concern "for considerable samples that overlap with 16S-sequenced ones, shotgun metagenomics which

rendered species-level classification, the authors should be able to combine these two datasets (based on compositional frequencies or patterns of temporal changes for example) and point out which species the highlighted ASVs are likely to be.” As we know, the two kinds of sequencing data cannot be directly compared due to the 16S rRNA genes only account for a small percentage of the whole genome. Therefore, there are very few genes found in 16S rRNA as in the shotgun metagenomics data.

We have now added the ASV table and taxonomy annotation to the Source Data file sheet “ASV table” as suggested. However, unlike OTU, according to the analysis methods, software and parameters provided in our methods section, anyone can get exactly the same ASV table from raw sequence data. This is why we did not provide the ASV table previously.

The comment on “*Also a minor point on Fig 2c: the annotations on the left column only appear for a few ASVs; do the other ASVs lack confident annotations (maybe not, as the main text discussed about ASV2513), or each annotation covers multiple ASVs?*” is well taken. We have now added a legend in Fig. 2c (line 752-753): “Genus annotations on the left column covers multiple right ASVs”.

The authors should also be cautious on their genus/family-level statements (Line 284-297), as all the analyses, as written, were done on the ASV (using 16S data) or species (using metagenomic data) level. For example, the increase of ASV2077 and ASV2513 alone is NOT equal to the increase of the whole Bacteroides genus, unless only two ASVs were found to be classified as Bacteroides (which seems unlikely). To support such statements on Bacteroides and Ruminococcaceae, the authors would need to perform LEfSe analyses on genus/family level (which is very feasible). If the authors instead decide to stick with ASV/species-level analyses, the conclusion should be modified to accurately reflect the data. In this case, again, having a biologically-meaningful taxonomy like Bacteroides uniformis would make the study much more referenceable than the current ASV1 etc.

Our response: As we know, the change of ASV is not equal to the functional change at the genus level, and it may even be the opposite of the ASV change in the same genus. Our genus level conclusion is based on that the ASVs members belonging to same genus (such as *Bacteroides* and *Ruminococcaceae*) changed consistently within genus. To further confirm the findings at the genus level, we have now performed a LEfSe analysis at the genus level as recommended. The new result has been added in the results section (line 166-169) as follows:

“To investigate whether changes of microbiome at the ASV level were similar to the changes at the genus level, we further performed a LEfSe analysis of the same parameters on the genus level. We found that those observed changed ASVs, if belonging to the same genus, indeed, they do have a similar trend at the genus level (Fig. S2).”

In addition, to make it clear, we have modified the “*Bacteroides/Ruminococcaceae*” in the discussion section to be as “members in *Bacteroides/Ruminococcaceae*” (line 287-303).

Regarding the reviewer’s concern “*In this case, again, having a biologically-meaningful taxonomy like Bacteroides uniformis would make the study much more referenceable than the current ASV1 etc.*” We believe that in the study of microbiome, currently it is more meaningful for many cases by using the ASV level than the taxonomy. For example, *E. coli* O157:H7 causes a severe intestinal infection in humans, while *E. coli* Nissle 1917 is a powerful probiotic strain. This is because the specific functions of microbiome are

determined by the strain level, and microbiome with the same taxonomic level may have completely opposite functions. The gut microbial ecosystem consists of bacterial populations as individual members, each of which has genetically identical cells derived from the same parent cell (D. E. Lea, *Journal of genetics* 1949, Dec, **49**, 264). Any attempts to dissect the contribution of the gut microbiome to human metabolic diseases starting with microbiome-wide association studies must recognize that the disease relevant functions of the gut microbiota may well be strain-specific. Strains/species in the same taxon from genus up to phylum can show widely diverse relationships with a particular disease phenotype - some may be positively associated, some negatively and others may not be associated at all. If a function is encoded in the “core-genome” of a taxon, all members of that taxon should have that function. If the function is encoded in the pan-genome, only one or a limited number of members would have that function. Therefore, we prefer to use ASV-level analysis and report as ASV1 in *Bacteroides uniformis* but not just *Bacteroides uniformis*.

REVIEWERS' COMMENTS

Reviewer #5 (Remarks to the Author):

I am happy that the authors have addressed my comments.

Reviewer #6 (Remarks to the Author):

All the previous comments regarding the statistics aspects have been adequately address. I don't have further comments.